# Quantifying Chilling Injury on the Photosynthesis System of Strawberries: Insights from Photosynthetic Fluorescence Characteristics and Hyperspectral Inversion

**DOI:** 10.3390/plants12173138

**Published:** 2023-08-31

**Authors:** Nan Jiang, Zaiqiang Yang, Jing Luo, Canyue Wang

**Affiliations:** School of Applied Meteorology, Nanjing University of Information Science & Technology, Nanjing 210044, China; 202211080002@nuist.edu.cn (N.J.);

**Keywords:** chlorophyll fluorescence, different photoperiod types, flowering and fruit-setting stage, *Fragaria × ananassa* Duch, gas exchange parameters, principal component analysis, SIF, spectral index, sustained low temperature, wavelet transform

## Abstract

Chilling injury can adversely affect strawberry bud differentiation, pollen vitality, fruit yield, and quality. Photosynthesis is a fundamental process that sustains plant life. However, different strawberry varieties exhibit varying levels of cold adaptability. Quantitatively evaluating the physiological activity of the photosynthetic system under low-temperature chilling injury remains a challenge. In this study, we investigated the effects of different levels of chilling stress on twenty photosynthetic fluorescence parameters in strawberry plants, using short-day strawberry variety “Toyonoka” and day-neutral variety “Selva” as representatives. Three dynamic chilling treatment levels (20/10 °C, 15/5 °C, and 10/0 °C) and three durations (3 days, 6 days, and 9 days) were applied to each variety. WUE, LCP, Y(II), qN, ${SIF}_{O_{2}-B}$ and ${rSIF}_{O_{2}-B}$ were selected as crucial indicators of strawberry photosynthetic physiological activity. Subsequently, we constructed a comprehensive score to assess the strawberry photosynthetic system under chilling injury and established a hyperspectral inversion model for stress quantification. The results indicate that the short-day strawberry “Toyonoka” exhibited a recovery effect under continuous 20/10 °C treatment, while the day-neutral variety “Selva” experienced progressively worsening stress levels across all temperature groups, with stress severity higher than that in “Toyonoka”. The BPNN model for the comprehensive assessment of the strawberry photosynthetic system under chilling injury showed optimal performance. It achieved a stress level prediction accuracy of 71.25% in 80 validation samples, with an R^2^ of 0.682 when fitted to actual results. This study provides scientific insights for the application of canopy remote sensing diagnostics of strawberry photosynthetic physiological chilling injury in practical agricultural production.

## 1. Introduction

Strawberry (*Fragaria × ananassa* Duch.) is known for its exquisite and delectable fruits, rich in beneficial substances such as vitamins, anthocyanins, flavonoids, minerals, and phenolic compounds, making it highly popular among consumers worldwide. It is considered one of the most important horticultural crops [1,2,3]. Based on differences in photoperiod sensitivity, strawberries are generally classified into short-day type and long-day type. Short-day varieties are sensitive to photoperiod and require no more than 12 h of daylight, leading to flowering and fruiting during the autumn and winter seasons, often referred to as “seasonal strawberries”. On the other hand, long-day strawberries require at least 12 h of daylight for proper bud differentiation and fruit production. These varieties can grow and produce fruits throughout the year under suitable temperatures, without a defined dormant period, earning them the name “everbearing strawberries” [4].

According to records, more than 95% of the global regions experience temperatures below 5 °C each year [5]. During the critical flowering and fruit-setting stage of strawberries (winter and early spring), the temperatures in major strawberry-producing regions, such as China and the United States, are often influenced by astronomical and meteorological factors, such as the distance from the sun, solar altitude, and monsoon, resulting in relatively low temperature levels [6]. Additionally, as global climate change intensifies, the frequency of extreme weather events, including cold extremes, is expected to rise significantly in the future [7]. Strawberries thrive best within a temperature range of 15–25 °C, and they are highly sensitive to low temperatures. Chilling injury (T ≥ 0 °C) during the flowering and fruit-setting stage has been shown to inhibit strawberry bud differentiation and gametophyte development, consequently negatively impacting fruit yield and quality [8,9,10].

Photosynthesis stands as the fundamental life process in plants, serving as their primary source of energy and organic matter [11]. It is a highly integrated process that is sensitive to changes in meteorological factors, and low-temperature conditions exacerbate the imbalance between the energy absorbed by the plant’s photosystem and the metabolic consumption, compelling plants to undergo photo-physical and photochemical adjustments to seek a new balance for sustained growth [12]. Parameters of photosynthesis (A, $g_{s}$, E, etc.) [13,14], dynamics of chlorophyll fluorescence ($F_{v}/F_{m}$, Y(Ⅱ), qP, etc.) [15,16], and solar-induced chlorophyll fluorescence (SIF) [17,18] are considered valuable indicators to reveal the physiological regulatory effects of the photosynthetic system under temperature stress. As a result, these photosynthetic fluorescence metrics are widely used as effective probes to assess plant photosynthetic activity [19].

In recent years, sensor technology has undergone rapid development, and hyperspectral techniques are becoming a crucial means for non-contact and quantitative assessments of crop physiological characteristics and nutritional status in precision agriculture [20,21]. Changes in leaf structure, pigment content, and elemental composition under different biological or abiotic factors can result in variations in the reflectance spectra of plants [22,23]. The Normalized Difference Vegetation Index (NDVI) is the most classic and widely used physiological evaluation indicator in plant science and remote sensing. It is calculated by normalizing the spectral reflectance in the near-infrared band (~850 nm, reflecting leaf structure) and the red band (~650 nm, reflecting pigment absorption). NDVI has been extensively employed for the physiological parameter inversion and modeling of crops. However, its susceptibility to saturation limits its application in diagnosing suboptimal plant health conditions [24,25]. Therefore, researchers have increasingly moved away from traditional vegetation indices, and have adopted statistical analysis methods to extract one or several spectral bands with stronger correlations to their research objectives from hyperspectral data. For instance, specialized spectral inversion models have been established to estimate net photosynthetic rate (A) [26], maximum photochemical efficiency ($F_{v}/F_{m}$) [27], and SIF [28]. However, these studies lack consistency and continuity in their research objectives, often focusing on single indicators for a specific plant species, thus failing to reflect the overall activity of the plant’s photosynthetic system. Hence, the establishment of a comprehensive hyperspectral assessment model capable of evaluating changes in the overall activity of the photosynthetic system in response to environmental variations holds great significance.

Jiang et al. previously revealed that under a 12 h constant nighttime low-temperature treatment, short-day strawberries exhibited a lower cold injury temperature threshold and demonstrated better low-temperature resistance compared to day-neutral strawberries [29]. However, like many experiments involving temperature control variables, such constant sub-zero temperature scenarios are rarely encountered in actual strawberry cultivation, which often involves protective measures such as greenhouses or plastic tunnels. Low-temperature events, such as cold waves, in major strawberry-producing regions globally typically last for no more than 10 days [30,31]. Inspired by prior research, we hypothesize that non-lethal daytime dynamic chilling exposures may have different effects on the photosynthetic system activity in different photoperiod types of strawberries. Single-type photosynthetic fluorescence indicators might not accurately represent the overall performance of photosynthetic system. Employing a combination of various photosynthetic evaluation indicators may provide a more comprehensive quantification of strawberry photosynthetic physiological activity. Moreover, to our knowledge, no studies have been reported in this context. Therefore, the aim of this study is to investigate the effects and differences of sustained dynamic chilling (≥0 °C) exposure during the flowering and fruit-setting stage on the photosynthetic fluorescence parameters and overall photosynthetic activity of short-day and long-day strawberries, and establish a hyperspectral quantitative inversion model for assessing the degree of chilling injury on the strawberry photosynthetic system.

## 2. Results

### 2.1. Impact of Continuous Dynamic Chilling Stress on Photosynthetic Parameters in Short-Day and Long-Day Strawberry Varieties

#### 2.1.1. Light Response Parameters

As the treatment temperature decreased and the duration increased, both the short-day strawberry variety “Toyonoka” and the long-day strawberry variety “Selva” exhibited a continuous decline in maximum net photosynthetic rate (A_max_) (Figure 1a,b). Following a 9-day (D9) exposure to 20 °C/10 °C (T1), “Toyonoka” showed a decrease of 15.6% in A_max_ compared to 25 °C/15 °C (CK), with no significant impact observed for different durations at this temperature. “Selva” displayed a similar decline of 15.7% in A_max_ after 9 days under T1, reaching 12.88 $\mu molm^{-2}s^{-1}$. Under 15 °C/5 °C (T2) treatment for 3 days (D3), both “Toyonoka” and “Selva” showed reductions of 2.00 and 3.38 $\mu molm^{-2}s^{-1}$, respectively, in A_max_ compared to CK. Under the T2 treatment for 6 days (D6), the A_max_ of “Selva” exhibited equivalence to that of “Toyonoka” treated for 6 days under 10 °C/0 °C (T3) conditions. Between 6 and 9 days of continuous treatment at T2 and T3, “Selva” exhibited a larger rate of A_max_ decline compared to “Toyonoka”. At T3D9 (continuously treated for 9 days under T3 temperature conditions, the same applies below.), “Toyonoka” demonstrated an A_max_ value of 8.60 $\mu molm^{-2}s^{-1}$, representing a 46.44% decrease compared to CK; meanwhile, “Selva” exhibited a substantial reduction of 4.64 $\mu molm^{-2}s^{-1}$, accounting for only 29.47% of CK.

In the T1D3 treatment, the short-day strawberry variety “Toyonoka” exhibited an increase of 3.20 $\mu molm^{-2}s^{-1}$ in light compensation point (LCP) (Figure 1c) compared to CK and a decrease of 99.15 $\mu molm^{-2}s^{-1}$ in light saturation point (LSP) (Figure 1e). Continuing the T1 treatment for 6 and 9 days resulted in an expansion of the light response range in “Toyonoka” compared to T1D3. Under the T1D9 treatment, “Toyonoka” showed a recovery of LCP to the CK level and a significant increase in LSP, reaching 1589.83 $\mu molm^{-2}s^{-1}$. Under T2 for 3 days, “Toyonoka” exhibited LCP and LSP levels similar to T1D3, with LCP continuously increasing as the duration extended, reaching 19.46 $\mu molm^{-2}s^{-1}$ at T2D9. LSP, after decreasing to levels similar to T1D3 under the T2D3 treatment, began to rise, but the magnitude of increase was significantly smaller than that observed in the T1 treatment. At T2D6, “Toyonoka” LSP recovered to the CK level and increased by 70.85 $\mu molm^{-2}s^{-1}$ after an additional 3-day treatment, although the difference was not significant. Under T3, “Toyonoka” LCP exhibited an accelerated increase with prolonged duration, with LCP at T3D6 being approximately twice that of CK, and reaching 25.28 $\mu molm^{-2}s^{-1}$ at T3D9. At this temperature, “Toyonoka” LSP drastically decreased to 794.57 $\mu molm^{-2}s^{-1}$ after 3 days of treatment and showed a slight increase as the treatment time progressed.

The LCP (Figure 1d) and LSP (Figure 1f) of the long-day strawberry variety “Selva” exhibited monotonic changes with decreasing temperature and prolonged duration. Under the T1D3 treatment, LCP showed a slight increase compared to CK. At T1D6, LCP was 5.62 $\mu molm^{-2}s^{-1}$ higher than T1D3 and 2.58 $\mu molm^{-2}s^{-1}$ lower than T1D9. After 3 days of treatment at T2, LCP increased by 7.90 $\mu molm^{-2}s^{-1}$ compared to CK and reached 2.64 times that of CK at T2D9. “Selva” exhibited strong sensitivity in LCP under the T3 treatment, with LCP increasing to 22.29 $\mu molm^{-2}s^{-1}$ at T3D3 and showing slight, nonsignificant increases from T3D6 to T3D9. The LSP of “Selva” displayed an approximately linear decreasing trend with decreasing temperature after 3 days of treatment at the three different chilling levels. The LSP of “Selva” was similar to that of T1D6 at T2D3 and T2D6; at T2D9, it was approximately 603.51 $\mu molm^{-2}s^{-1}$, similar to T3D6. However, at T3D9, “Selva” showed a severely reduced LSP of only 488.50 $\mu molm^{-2}s^{-1}$, indicating significant photoinhibition.

#### 2.1.2. Gas Exchange Parameters

During the early stages of chilling stress, the stomatal conductance ($g_{s}$) of the short-day strawberry variety “Toyonoka” (Figure 2a) showed a milder decline compared to the long-day strawberry variety “Selva” (Figure 2b) under the three different temperature gradients (T1~T3). Under T1D3 treatment, the $g_{s}$ of “Toyonoka” was comparable to that of T2D3, and it exhibited a slow, approximately linear decline at T1D6 and T1D9. After 6 and 9 days of T2 treatment, “Toyonoka” $g_{s}$ declined by a magnitude similar to the later stages of continuous T1 treatment, but remained consistently around 0.87 $mmolm^{-2}s^{-1}$ lower than T1 at the same treatment duration. At T3D3, “Toyonoka” $g_{s}$ was comparable to that of T1D6, while at T3D6, it resembled the level of T2D6. At T3D9, the “Toyonoka” $g_{s}$ value decreased to 0.082 $mmolm^{-2}s^{-1}$, representing a 50.12% decrease compared to CK. In the early stages of low-temperature stress at the three different levels, “Selva” exhibited a significant decrease in $g_{s}$. Between T1D6 and T1D9, “Selva” showed relatively stable $g_{s}$, while under T2 and T3 treatments, the decline in $g_{s}$ from 6 to 9 days was greater than that observed during the initial 3 to 6 days at the same temperature. Additionally, at T3D6, “Selva” $g_{s}$ was 0.069 $mmolm^{-2}s^{-1}$, not significantly different from T3D3 and similar to the effect observed at T2D9. At T3D9, “Selva” $g_{s}$ decreased to 0.0414 $mmolm^{-2}s^{-1}$, representing a 74.93% decrease compared to CK, approximately half of that observed in “Toyonoka” under the same treatment.

The transpiration rate (E) of the short-day strawberry variety “Toyonoka” and the long-day strawberry variety “Selva” (Figure 2c,d) exhibited significant differences in consistency under different levels of continuous chilling stress for 9 days. “Toyonoka” maintained E levels ranging from 1.76 to 2.29 $mmolm^{-2}s^{-1}$ after 9 days of continuous treatment at T1, T2, and T3, with no significant differences observed. In contrast, “Selva” displayed a decreasing trend in E with decreasing treatment temperatures over 9 days, with values of 1.98, 1.59, and 1.28 $mmolm^{-2}s^{-1}$ at T1D9, T2D9, and T3D9, respectively, and T1D9 being comparable to T1D6. At T3D3, “Selva” exhibited a 40.33% decrease in E compared to CK, a larger reduction than the 26.55% observed in “Toyonoka” compared to CK. However, both “Toyonoka” and “Selva” demonstrated stabilizing characteristics at T3D6 and T3D9. In summary, the short-day strawberry variety “Toyonoka” showed a more rapid decline in E to lower levels than the long-day strawberry variety “Selva” under chilling conditions.

As shown in Figure 2e,f, the ratio of intercellular ${CO}_{2}$ concentration to atmospheric ${CO}_{2}$ concentration (ci/ca) of the short-day strawberry variety “Toyonoka” and the long-day strawberry variety “Selva” was consistently lower than CK under different time treatments at T1. Under T2, “Toyonoka” displayed an initial decrease followed by a subsequent increase in ci/ca with prolonged duration, but it remained lower than both T1 and CK. Under the T2 treatment, “Selva” exhibited similar ci/ca levels at 3 and 6 days of treatment, while at T2D9, it showed a greater increase compared to T2D6, reaching 1.59, which is slightly higher than CK. Under T3, the ci/ca of “Toyonoka” showed no significant differences at different durations, with a relatively higher value of 0.61 observed at T3D9, representing a 5.59% increase compared to CK. “Selva” displayed a gradual increase in ci/ca within the T3D3 and T3D6 range, both of which were higher than CK. At T3D9, “Selva” exhibited a larger increase in ci/ca, reaching 0.66.

#### 2.1.3. Parameters of Water and Light Use Efficiency

The water use efficiency (WUE) of the short-day strawberry variety “Toyonoka” (Figure 3a) showed no significant changes compared to CK after 3 days of treatment at different levels of chilling stress. Additionally, the impact on WUE remained positive after 6 days of continuous treatment at different temperatures, with no significant differences observed compared to CK and an average increase of 13.95% relative to CK. Under the T1D9 treatment, “Toyonoka” WUE continued to increase, reaching 7.39 $\mu mol{mmol}^{-1}$, a 46.30% increase compared to CK. At T2D9, “Toyonoka” WUE levels were similar to those of T2D6. After T3D9 treatment, the “Toyonoka” WUE value returned to CK levels. For the long-day strawberry variety “Selva” (Figure 3b), the effects on WUE during the first 6 days of treatment at different levels of temperature were similar to those seen in “Toyonoka”, except for a significantly higher increase in WUE at T3D3, reaching 6.62 $\mu mol{mmol}^{-1}$, compared to other temperature treatment groups. Under T1D9 treatment, “Selva” WUE did not continue to increase like “Toyonoka”, but remained at levels similar to T1D6, which were also close to the WUE level at T3D3. “Selva” WUE at T2D9 reverted to CK levels and drastically decreased to 3.51 $\mu mol{mmol}^{-1}$ after the T3D9 treatment, representing a 39.94% reduction compared to CK.

The apparent quantum efficiency (AQE) characterizes the efficiency of the photosynthetic conversion of light energy into biomass energy in plants. This parameter revealed contrasting response effects in the light energy utilization ability between the short-day strawberry variety “Toyonoka” and the long-day strawberry variety “Selva” at T1 (Figure 3c,d). Under T1D3 treatment, “Toyonoka” AQE decreased by 24.25% compared to CK, with no significant differences observed between T2D3 and T3D3. However, with increasing duration, AQE gradually increased, reaching CK levels at T1D9. AQE stabilized at T2D6 and T2D9, with an average reduction of 0.007 $\mu mol{CO}_{2}{\mu mol}^{-1}photons$ compared to T2D3. “Toyonoka” AQE exhibited the largest decrease during the first 3 days under T3 treatment, with a reduction of 0.030 $\mu mol{CO}_{2}{\mu mol}^{-1}photons$, followed by a stable decline as treatment time extended, reaching 0.031 $\mu mol{CO}_{2}{\mu mol}^{-1}photons$ at T3D9. In contrast, “Selva” AQE decreased to 0.039 $\mu mol{CO}_{2}{\mu mol}^{-1}photons$ at T1D6, similar to the level observed at T2D3. Under T1D9 treatment, “Selva” AQE decreased by 43.79% compared to CK, similar to the reduction observed in “Toyonoka” at T3D9 compared to CK. At T3D3, “Selva” AQE significantly decreased by 50.97% compared to CK, with further decreases observed with prolonged treatment under T3, reaching 0.020 $\mu mol{CO}_{2}{\mu mol}^{-1}photons$ at T3D9, only 32.67% of CK.

### 2.2. Impact of Continuous Dynamic Chilling Stress on Chlorophyll Fluorescence Induction Kinetic Parameters in Short-Day and Long-Day Strawberry Varieties

#### 2.2.1. Photosystem II Photochemical Efficiency Parameters

The trends of maximum photosystem II (PSII) photochemical efficiency (Fv/Fm) and actual PSII photochemical efficiency (${Fv}^{\prime }/{Fm}^{\prime }$) in the short-day strawberry variety “Toyonoka” and the long-day strawberry variety “Selva” were similar under different durations of T1 and T2 treatments (Figure 4). “Toyonoka” exhibited a decreasing and then increasing trend in PSII photochemical efficiency under T1 treatment, reaching a turning point after 6 days of continuous treatment and returning to the level of T1D3 after 9 days of treatment. “Selva” showed stable photosystem II photochemical efficiency near CK levels at various durations of T1 treatment, without significant fluctuations. Under T2 treatment, “Toyonoka” Fv/Fm and ${Fv}^{\prime }/{Fm}^{\prime }$ continued to decrease within the first 6 days of continuous treatment and then stabilized at T2D6 and T2D9, with reductions of 6.07% and 6.48%, respectively, compared to CK. “Selva” exhibited a significant decrease in PSII photochemical efficiency within the first 3 days of treatment at this temperature, but the values of Fv/Fm and ${Fv}^{\prime }/{Fm}^{\prime }$ remained stable with treatment time at 3, 6, and 9 days, reaching 95.78% and 95.72% of CK, respectively, at T2D9. At T3, the changes in Fv/Fm and ${Fv}^{\prime }/{Fm}^{\prime }$ of both strawberry varieties differed with an increasing duration of chilling treatment. “Toyonoka” showed a decreasing trend in Fv/Fm followed by a stable phase under T3 treatment, reaching 0.76 at T3D9, which was 91.83% of CK. On the other hand, ${Fv}^{\prime }/{Fm}^{\prime }$ continuously decreased with prolonged treatment, showing reductions of 0.001, 0.015, and 0.035 at 3, 6, and 9 days, respectively. “Selva” exhibited a slight decline in Fv/Fm under T3 treatment with increasing stress duration, while ${Fv}^{\prime }/{Fm}^{\prime }$ remained stable at 3, 6, and 9 days, being 0.014, 0.007, and 0.001 lower than Fv/Fm, respectively.

#### 2.2.2. Fluorescence Quenching Coefficients

Under T1 treatment, both the short-day strawberry variety “Toyonoka” and the long-day strawberry variety “Selva” exhibited stable and consistently higher lake-type non-photochemical quenching coefficients (qL) compared to CK, with an average increase of 0.101 (Figure 5a,b). In the case of “Toyonoka”, qL was similar to T1D3 when treated under T2D3, but with prolonged treatment, it decreased to a level close to CK at T2D9. At T3D3, “Toyonoka” showed a decrease of 18.41% in qL compared to CK, and maintained stability in the range of 0.477 to 0.485 as the treatment duration increased. On the other hand, “Selva” exhibited a qL decrease of 4.93% compared to CK under T2D3, followed by a slight decline of 0.024 after 6 and 9 days of continuous treatment, reaching 0.505, but this was still lower than CK (0.538). Under T3D3 treatment, “Selva” qL decreased to 0.455. At T3D6 and T3D9, “Selva” showed stable qL values, with reductions of 23.52% and 22.37% compared to CK, respectively.

Under T1 treatment, the non-photochemical quenching coefficient (qN) in the short-day strawberry variety “Toyonoka” exhibited an initial increase followed by a decrease with prolonged treatment duration (Figure 5c). At T1D3, “Toyonoka” qN was similar to CK, while it increased to 0.037 at T1D6 and returned to T1D3 levels at T1D9, with no significant difference from CK. Under T2 treatment, “Toyonoka” qN was 13.31% higher than CK at T2D3, further increasing to 0.039 after 6 days, similar to T3D3 levels, and then declining to the same level as T2D3 at T2D9. Under T3 stress, qN showed a substantial increase of 1.28-fold compared to CK in the initial 3 days, and then remained stable with a slight upward trend as the treatment duration extended. In the long-day strawberry variety “Selva”, qN (Figure 5d) at T1D6 was close to T1D3 at 0.034 and increased to 0.038 at T1D9, representing a 29.90% increase compared to CK. At T2D9, “Selva” qN increased to 0.0394, similar to “Toyonoka” qN at T3D6. Under T3D3, “Selva” qN reached 1.51 times that of CK, and it remained stable around 0.422 after 6 and 9 days of continuous treatment.

The discrepancy between 1 and qP delineates the redox state of Q_A_, representing the fraction of PSII reaction centers in the non-photochemical quenched state [32]. Under T1 treatment, both the short-day strawberry variety “Toyonoka” and the long-day strawberry variety “Selva” exhibited lower 1−qP values compared to CK at different durations (Figure 5e,f). For “Toyonoka”, the 1−qP at T2, lasting for 3 days, decreased by 0.037 compared to CK, which was similar to the level observed at T1D3. At T1D6, it rebounded to the CK level, and at T2D9, it was 18.09% higher than CK. Under T3 treatment, “Toyonoka” showed no significant difference in 1−qP compared to CK, but with an increase in duration, the 1−qP values at T3D6 and T3D9 were 25.61% and 28.79% higher than CK, respectively. On the other hand, “Selva” exhibited high sensitivity to T2 and T3 treatments, reaching stable 1−qP levels at T2D3 and T3D3, which were 0.140 and 0.148, respectively, and did not show significant differences compared to the 1−qP values after 6 and 9 days of treatment.

#### 2.2.3. Photosystem II Quantum Yield Parameters

The PSⅡ quantum yield (Y(Ⅱ)) of the short-day strawberry variety “Toyonoka” (Figure 6a) exhibited a decreasing trend from T1, lasting for 3 to 9 days, followed by an increase. In contrast, the long-day strawberry variety “Selva” showed an increase in Y(Ⅱ) until day 6, after which it stabilized (Figure 6b). Under T2 treatment, “Toyonoka” showed a 3.25% increase in Y(Ⅱ) compared to CK at day 3, similar to the level observed at T1D3. However, at T2D6, Y(Ⅱ) significantly decreased, and after T2D9 treatment, it further dropped to 0.659. In contrast, “Selva” exhibited a 6.3% decrease in Y(Ⅱ) compared to CK at day 3 of T2, but it significantly increased to 0.706 at T2D6 and then declined to 0.670 with prolonged duration. With increasing treatment time, “Toyonoka” displayed a near-linear and steady decline in Y(Ⅱ) at T3, reaching 0.637 at T3D9, which was 11.83% lower than CK. For “Selva”, after 3 days of treatment at T3, Y(Ⅱ) decreased to 0.667, which is 93.43% of the value of CK, and it remained stable at the same level after 6 and 9 days of continuous treatment.

Under T1D3 treatment, the PSII regulatory energy dissipation quantum yield (Y(NPQ)) of the short-day strawberry variety “Toyonoka” (Figure 6c) was similar to CK. It increased to 0.0089 at T1D6 and then decreased to 0.0072 at T1D9, consistently higher than the CK average level of 0.0066. “Toyonoka” exhibited a slight increase in Y(NPQ) compared to CK at T2D3, and it further increased to 0.0103 at T2D6, which is 1.55 times higher than CK, and it remained stable at this level at T2D9. Y(NPQ) continuously increased under T3 treatment with prolonged duration, reaching the same level as T1D6 at T3D3 and 0.0124 at T3D9, approximately twice the value of CK. For the long-day strawberry variety “Selva”, under different degrees of chilling treatment for 3 days, the increase in Y(NPQ) (Figure 6d) was greater than that of “Toyonoka”. However, at T1 and T2 with continuous treatment for 6 days, Y(NPQ) remained at the same level as that of the 3-day treatment. At T1D9 and T2D9, Y(NPQ) increased again, reaching 0.0083 and 0.0102, respectively. Under T3 treatment, the trend of Y(NPQ) was different from those of T1 and T2. At T3D6, it was similar to T3D3, and at T3D9, it decreased to 0.0088.

The PSII non-regulatory energy dissipation quantum yield (Y(NO)) of the short-day strawberry variety “Toyonoka” (Figure 6e) exhibited a fluctuating pattern with the duration of T1 treatment. It decreased by 7.84% compared to CK at T1D3, then increased by 7.12% at T1D6, and finally returned to the CK level at T1D9. In contrast, the Y(NO) of the long-day strawberry variety “Selva” (Figure 6f) remained relatively stable after the T1 treatment for 6 and 9 days, slightly lower than T1D3, and slightly lower than CK. Under T2 treatment, “Toyonoka” showed Y(NO) levels similar to T1D3 after 3 days of stress, and it increased to 0.331 at T2D9, which was 22.00% higher than CK. “Toyonoka” exhibited Y(NO) levels similar to T2D6 under T3D3, then similar to T2D9 at T3D6, and reached the highest value at T3D9, which was 0.353. For “Selva” under T2 and T3 treatments, the Y(NO) remained relatively stable and did not show significant changes with the duration of treatment. It reached a stable point within the first 3 days of treatment, with values of 0.3 and 0.325 at T2D9 and T3D9, respectively, representing an 8.89% and 17.88% increase compared to CK.

### 2.3. Impact of Continuous Dynamic Chilling Stress on Solar-Induced Chlorophyll Fluorescence in Short-Day and Long-Day Strawberry Varieties

#### 2.3.1. Solar-Induced Chlorophyll Fluorescence 

Figure 7 shows that solar-induced chlorophyll fluorescence (SIF) retrieved based on the O_2_-B absorption band (${SIF}_{O_{2}-B}$) and the O_2_-A absorption band (${SIF}_{O_{2}-A}$) exhibited similar trends under various degrees of low-temperature and short-duration treatments, but the ${SIF}_{O_{2}-A}$ values were consistently higher than ${SIF}_{O_{2}-B}$. The short-day strawberry variety “Toyonoka” exhibited an average decrease of 18.95% compared to CK under T1D3 treatment, with a slight recovery after 6 days of continuous treatment, followed by a further decline. The SIF trend under T2 treatment was similar to that under T1, and after 9 days of low-temperature treatment, it decreased to the same level as T3D9. Under T3D3 treatment, “Toyonoka” showed an average decrease of 41.43% in SIF compared to CK. ${SIF}_{O_{2}-B}$ and ${SIF}_{O_{2}-A}$ further decreased and stabilized at 0.198–0.204 $mWm^{-2}{sr}^{-1}{nm}^{-1}$ and 0.247–0.269 $mWm^{-2}{sr}^{-1}{nm}^{-1}$, respectively, under T3D6 and T3D9. For the long-day strawberry variety “Selva”, SIF decreased continuously with the increase in treatment duration under T1 and stabilized after 6 days of T2 treatment. Both T1D9 and T2D9 showed SIF at the same level. “Selva” maintained a stable SIF under T3D3 and T3D6, averaging about 68.06% of CK. However, a decrease in SIF was observed again under T3D9, with SIF values of 0.145 and 0.154 $mWm^{-2}{sr}^{-1}{nm}^{-1}$ for O_2_-B and O_2_-A bands, respectively.

#### 2.3.2. Relative Solar-Induced Chlorophyll Fluorescence

The ratio of SIF to incident solar radiation, rSIF, has been considered to mitigate the influence of different light intensities on SIF (Figure 8) [33]. For the short-day strawberry variety “Toyonoka”, rSIF under T1 treatment initially decreased and then increased with the increase in treatment duration, stabilizing around the CK level. However, under T2 and T3 treatments, rSIF exhibited a continuous decrease. The magnitude of the decrease in rSIF from T3D6 to T3D9 was significantly smaller than that observed in the first 6 days of treatment, indicating a trend towards stabilization. Under T2D9 and T3D9, ${rSIF}_{O_{2}-B}$ and ${rSIF}_{O_{2}-A}$ were reduced by 34.36% and 47.06%, respectively, compared to their respective CK levels. As for the long-day strawberry variety “Selva”, the rSIF under T1 treatment decreased continuously with the increase in treatment duration. Under T1D3, rSIF was similar to that of “Toyonoka” under the same treatment. However, at T1D6, rSIF was comparable to “Toyonoka” under T2D6 and its own level at T2D3, and at T1D9, it decreased to 60.52% of CK. Under T2 treatment, “Selva” exhibited the largest decrease in rSIF during the first 3 days, with reductions of 0.004 and 0.020 in the O_2_-B and O_2_-A bands, respectively. The magnitude of the decrease gradually diminished with continuous treatment for 6 and 9 days. Under T3D3, “Selva” showed an average decrease of 47.22% compared to CK. The rate of decline between T3D3 and T3D6 slowed down, and rSIF significantly decreased again at T3D9. The average rSIF in both absorption bands was about 33.01% of CK.

### 2.4. Construction of a Comprehensive Evaluation System for Chilling Stress on the Photosynthetic System of Strawberry with Different Photoperiod Types

#### 2.4.1. Extraction of Photosynthetic Physiological Characteristic Indices

To achieve a comprehensive assessment of the overall photosynthetic system activity in strawberries with different photoperiod types, we conducted principal component analysis (PCA) on the eight photosynthetic parameters, eight chlorophyll fluorescence induction kinetic parameters, and four daylight-induced chlorophyll fluorescence parameters, as introduced earlier. This process aimed to extract characteristic indices that reflect various aspects of plant photosynthetic physiological performance, thereby reducing data redundancy. The PCA scores and loadings for different types of photosynthetic physiological parameters are illustrated in Figure 9.

The cumulative contribution of the first principal component (PC1) and the second principal component (PC2) to the variation in photosynthetic parameters amounted to 78.44%, signifying their substantial representation of the original data’s variability. Under the CK condition, the overlap of the 95% confidence intervals for photosynthetic parameters in “Toyonoka” and “Selva” was higher compared to the chilling treatment group, suggesting different response levels of photosynthetic parameters to the low-temperature environment among strawberries with different photoperiod types. Among the photosynthetic parameters, $A_{max}$, $g_{s}$, E, AQE, and LSP showed significant positive contributions to PC1, with $A_{max}$ exhibiting the highest positive loading of 0.417. On the other hand, LCP demonstrated a negative contribution to PC1, with an absolute loading value of 0.424, greater than that of $A_{max}$. Regarding PC2, WUE exhibited a substantial loading of 0.738, significantly higher than the other photosynthetic parameters.

The cumulative variance contribution of PC1 and PC2 for chlorophyll fluorescence induction kinetic parameters was 81.09%. Among them, the absolute loading of Y(Ⅱ) on PC1 was the highest, at 0.41, showing a similar direction and level to Fv/Fm and ${Fv}^{\prime }/{Fm}^{\prime }$, but a tendency towards negative correlation with the loading value of Y(NO). The contributions of the three fluorescence quenching coefficients to PC2 were higher than other fluorescence parameters, with qN having a positive loading of 0.527 on PC2, which was 0.029 higher than qL and 0.053 higher than the absolute value of 1−qP. The PC scores of “Toyonoka” and “Selva” showed a high degree of overlap in the 95% confidence intervals under CK and T3 treatments. The confidence intervals of “Selva” were encompassed within those of “Toyonoka” under T1 and T2 treatments. “Toyonoka” exhibited greater variability in PC scores under different durations compared to “Selva”.

PC1 of solar-induced chlorophyll fluorescence provided 84.11% of the information and, together with PC2, revealed 92.28% of the variability in the homogenous dataset. The SIF and rSIF extracted based on O_2_-B and O_2_-A atmospheric absorption bands showed positive contributions to PC1. In contrast, their contributions to PC2 were negative for SIF and positive for rSIF. Among them, ${SIF}_{O_{2}-B}$ and $r{SIF}_{O_{2}-B}$ exhibited the highest loadings on PC1 and PC2, with values of 0.507 and 0.696, respectively. Furthermore, from the graph, it is evident that the confidence intervals of solar-induced chlorophyll fluorescence parameters for the “Selva” under T2 treatment and “Toyonoka” under T3 treatment significantly overlapped, indicating similar levels of SIF and rSIF for these two strawberry varieties under these respective temperature treatments.

Based on the comprehensive analysis mentioned above, we identified the parameters making the highest contributions to each PC1 and PC2, namely: WUE, LCP, Y(Ⅱ), qN, ${SIF}_{O_{2}-B}$ and $r{SIF}_{O_{2}-B}$. These selected parameters serve as the primary characteristic indices among all 20 variables. Subsequently, we further analyzed these indices to construct a comprehensive assessment score for chilling injury on the photosynthetic system of strawberries.

#### 2.4.2. Construction and Grading of a Comprehensive Score for Photosynthetic System Chilling Injury in Strawberries

Photosynthesis is a complex process, and the information on plant photosynthetic system activity reflected by different types of photosynthetic physiological parameters may not be mutually independent, but rather coherently and dynamically interconnected. Therefore, we performed a second round of PCA on the six extracted photosynthetic physiological characteristic indices. This comprehensive analysis allowed us to examine the information contributed by different indices in various dimensions and compress similar or redundant information. The results are presented in Table 1. The cumulative variance contribution of the first three PCs exceeded 85%, indicating a robust capture of the primary features and patterns of the six characteristic indices. Based on the variance contribution and loadings provided by PC1 to PC3 in Table 1, we computed the PC scores for each dimension and utilized these scores to construct the Comprehensive Score for Photosynthetic System Chilling Injury (CSPC) in strawberries under chilling stress. The calculation formula is presented in Table 2.

In practical agricultural production, employing clear grades for disaster assessment and risk prediction is more common. Therefore, in this study, we analyzed a total of 432 sets of CSPC for both the short-day variety “Toyonoka” and the long-day cultivar “Selva”. We discovered that using the ‘SD/2’ interval as the grade threshold effectively indicates the variations in plant photosynthetic system activity under different levels of low-temperature stress. It also reflects the differences in chilling resistance and tolerance among strawberries with different photoperiod types. Therefore, starting with the average CSPC score of 0 for all chilling stress levels, we divided the scores into six grades, ranging from 0 to 5, with each grade threshold set at 0.54, as shown in Table 3. As the numerical value increases, it indicates a gradual increase in the chilling stress pressure experienced by the strawberry plants.

Based on the collected data of all photosynthetic physiological characteristic indices from experiments, we calculated the stress scores and stress levels of the photosynthetic systems in strawberries with different photoperiod types under varying degrees and durations of chilling stress. The results for each treatment group are presented in Table 4. From the results, it is evident that under the same temperature of chilling treatment, the long-day strawberry “Selva” experiences physiological stress earlier than the short-day strawberry “Toyonoka” in their photosynthetic system. Additionally, under the same duration of low-temperature stress, the CSPC of “Toyonoka” is generally higher than that of “Selva”, indicating that the photosynthetic system of short-day strawberries exhibits better resistance to chilling stress compared to the long-day variety.

### 2.5. Hyperspectral Inversion of Comprehensive Score for Photosynthetic System Chilling Injury in Strawberries with Different Photoperiod Types

#### 2.5.1. Hyperspectral Transformation Results

After transforming the raw spectra through mean filtering, envelope removal, and first-order differentiation to capture distinct spectral features of the hyperspectral data, the three categories of original spectral data were each subjected to a six-layer wavelet transformation, resulting in wavelet coefficient spectra. The Pearson correlation coefficients between the original spectra and wavelet coefficient spectra at different scales, along with the comprehensive scores for photosynthetic system chilling injury in strawberries, were illustrated in Figure 10. While retaining the original spectra prior to wavelet transformation for each type of spectrum, the wavelet transformation yielded six distinct sets of spectral information at varying scales for each spectral dataset. Noteworthy variations in the correlation with strawberry CSPC were observed among different types and scales of spectra across different wavelength bands.

For the original spectra that have not undergone wavelet transformation, spectra subjected solely to mean filtering exhibited a positive correlation across the entire wavelength range with strawberry CSPC, though the overall correlation remained moderate. Following envelope removal, the spectra displayed a more concentrated relationship with both positive and negative extremes of low-temperature damage stress scores at various wavelengths, highlighting more pronounced distinctions. As illustrated notably in the figure, the strongest positive correlation appeared within the 581–602 nm interval, while the most prominent negative correlation, characterized by the highest absolute correlation coefficient, was evident within the 620–640 nm range. Upon applying first-order differentiation to the original spectra subsequent to smooth filtering, the correlation coefficient spectra with low-temperature injury scores revealed weak associations spanning over half of the wavelength range, with relatively stronger band correlations observed only in the 700–727 nm and 870–900 nm regions.

After wavelet transformation and subsequent reconstruction of the high-frequency components across six scales for the three spectral types, certain transform scales exhibited correlation coefficients between spectral coefficients and CSPC exceeding those of the original spectra. Notably, such an enhancement was observed for scales like the mean-filtered CD1, CD2, and CD4, indicating that wavelet transformation has extracted spectral information more closely related to chilling injury stress severity from the original spectra. The correlation of the coefficients of the mean-filtered spectra CD1 and CD2 with CSPC was generally low across most wavelength bands, except for a narrow high-correlation interval at 876–892 nm. This phenomenon was similarly observed in the first- and second-level high-frequency components of the envelope-removed spectra and the first-order differential spectra. The wavelet coefficients derived from envelope-removed spectra across various scales exhibited an overall higher correlation with CSPC compared to the high-frequency components from mean filtering and first-order differentiation, with the first-order differential wavelet components displaying the largest low-correlation regions, such as 410–685 nm, 740–844 nm, and 920–990 nm. Additionally, the correlation coefficients for different spectral types in the sixth-scale high-frequency components were consistently lower than those of CD1–CD5, indicating that wavelet high-frequency components of the sixth or higher order no longer provide further effective feature information extraction for spectral data with a resolution of 2.6 nm and 224 bands.

#### 2.5.2. Extraction of Spectral Features for Photosynthetic System Chilling Injury

Prominent wavelength information exhibiting the strongest positive and negative correlations on the original spectra and high-frequency components across various scales for distinct spectral types was extracted. Subsequently, a simple combination of spectral indices was conducted on these two characteristic bands to further uncover the correlation between each transformation scheme’s feature spectral indices and CSPC. The results are presented in Table 5. In the table, among the 21 spectral transformation schemes, a total of six sets of feature spectral indices display maximum correlation coefficients exceeding 0.5. These are: the Ba (Band a) of mean-filtered CD1, CD2, and CD4 spectra (with respective coefficients of 0.534, 0.510, and 0.504); the Ba of the envelope-removed original spectrum (0.516) and the difference between Ba and Bb (Ba − Bb) of CD1 (0.521); and the Ba − Bb of first-order differential spectrum CD5 (0.532). However, operations involving the ratio and normalization of Ba and Bb did not significantly enhance the correlation coefficients. In conclusion, these six indices were selected as spectral features indicative of photosynthetic system chilling injury, serving as pivotal input data for constructing a hyperspectral inversion model targeting the severity of chilling injury in the strawberry photosynthetic system.

#### 2.5.3. Hyperspectral Inversion Modeling and Accuracy Evaluation of Photosynthetic System Chilling Injury Severity in Strawberries

To contrast the impact of different modeling approaches on the accuracy of hyperspectral inversion for photosynthetic system chilling injury severity in strawberries, and to select the most appropriate modeling method, this study employed four classical learning algorithms: Random Forest (RF), Backpropagation Neural Network (BPNN), Support Vector Regression (SVR), and Ridge Regression (RR). To conduct this comparative analysis, a total of 432 sets of spectral features and CSPC data were randomly processed. Among these, 352 sets were utilized as model training data, with the remaining 80 sets used for accuracy validation. The validation outcomes are depicted in Figure 11. The predictions of CSPC by RF, BPNN, and SVR displayed favorable accuracy in the validation results. The coefficients of determination (R^2^) and root mean square errors (RMSE) were 0.706, 0.682, 0.682 and 0.538, 0.571, 0.751, respectively. The predictive accuracy of RF slightly surpassed that of BPNN and SVR, with RF’s predictions showing a better fitting accuracy. However, BPNN’s predictions exhibited a fitting equation closer to a 1:1 line with actual CSPC values. Conversely, predictions based on RR were concentrated near zero, indicating poorer fitting accuracy with R^2^ and RMSE values reflecting significant deviation from the true values. Consequently, RR’s predictions were unsuitable for this modeling application.

The predicted strawberry CSPC values from the 80 validation samples, obtained using different models, were transformed into stress levels. The similarities and discrepancies between true-predicted differences and predictions from various models are illustrated in Figure 12. Due to the higher tolerance of stress levels towards prediction deviations, the hyperspectral inversion accuracy of strawberry chilling injury severity based on BPNN outperformed RF and SVR by 12.5% and 21.25%, respectively. Moreover, the number of correctly predicted samples using BPNN exceeded those of RF and SVR by 10 and 17 samples, respectively. In contrast, the number of correctly predicted results using RR amounted to only 16, suggesting a lack of diagnostic capability for low-temperature stress severity levels.

Collectively, the results indicate that RF and BPNN have demonstrated outstanding performance in establishing hyperspectral inversion models for strawberry photosynthetic system chilling injury severity levels. This also underscores the strong applicability of ensemble learning and deep learning methods, showcasing their efficacy in unearthing intricate relationships embedded within complex data. In contrast, the predictive outcomes of SVR were less stable than those of RF and BPNN, displaying significant deviations in the predictions. On the other hand, the linear regression learning model, RR, was found to be inadequate for complex hyperspectral inversion modeling.

## 3. Discussion

The factors contributing to meteorological disasters encompass the causative agents, the vulnerability of the affected entities, and their exposure [34]. Photosynthesis stands as the cornerstone of plant vitality, and the flowering and fruiting phase notably constitutes the most pivotal growth stage within strawberry agriculture [35]. Suboptimal temperatures not only impede pollen and ovule development, reducing pollination success and fruit set, but also escalate the plants’ nutrient consumption for survival maintenance. This, in turn, diminishes nutrient allocation to the fruits, thereby compromising yield and quality [36]. Hence, this investigation innovatively dissects the multifaceted photosynthetic physiological activities during varying photoperiodic strawberry growth phases, elucidating the impacts of diverse degrees and durations of supra-zero temperatures on light response range, gas exchange proficiency, photochemical reaction efficiency, photothermal self-protective mechanisms, and steady-state fluorescence. By integrating these insights, a novel assessment framework, rooted in photosynthetic fluorescence perspectives, quantifying the extent of physiological stress imposed by chilling adversity on different photoperiodic strawberry varieties, is formulated. Leveraging this foundation, a hyperspectral inversion model is established, enabling swift, quantitative, and non-intrusive diagnostics of chilling stress on the photosynthetic system. These findings hold promise in furnishing the strawberry cultivation sector with scientific underpinnings for preemptive measures and evaluation against cold stress.

In our investigation, we observed that the short-day strawberry cultivar “Toyonoka” exhibited a reduction in its initial 3-day light response range under 20/10 °C treatment compared to CK. However, with the progression of the treatment duration, an expanding trend was noted, denoting a decrease in LCP and an elevation in LSP. This phenomenon could be attributed to the impact of early exposure to cold temperatures on the activity of photosynthesis-related enzymes, photosynthetic pigment content, and functionality. As a result, the plants require a higher light intensity to sustain fundamental photosynthetic levels, representing a stress response to unfavorable conditions. Yet, as the duration of cold treatment extended, the cold-resistant “Toyonoka” cultivar adapted to the environment, developing novel strategies such as protective enzymes and antioxidant substances to acclimate to the new surroundings. Consequently, it gradually regained the ability to perform photosynthesis under lower light intensities. These findings indicate that “Toyonoka” possesses this regulatory capability at both 20/10 °C and 15/5 °C, whereas the long-day strawberry cultivar “Selva” did not exhibit similar self-adjusting effects to cold conditions. Stomata serve as conduits for carbon and water exchange between plants and the atmosphere, their movements reflecting the plant’s metabolic status, thus serving as crucial indicators of chilling stress [37]. Under cold conditions, plants typically reduce or close their stomata to mitigate transpiration and water loss, preventing excessive dehydration and cold-induced damage to leaves. The decline rate of $g_{s}$ was more pronounced in the long-day “Selva” cultivar than in the short-day “Toyonoka” cultivar under escalating chilling stress. However, concerning E, “Toyonoka” exhibited a more aggressive decline at 20/10 °C and 15/5 °C compared to “Selva”. Given the significantly higher increase in WUE with prolonged cold exposure in “Toyonoka” at 20/10 °C and 15/5 °C, as opposed to “Selva”, it is conceivable that “Toyonoka” employs metabolic regulatory pathways to suppress E, potentially involving adjustments in water transport routes, hormone levels, and signal transduction pathways [38].

Under continuous exposure to 10/0 °C for 6 and 9 days, a significant increase in the difference between Fv/Fm and ${Fv}^{\prime }/{Fm}^{\prime }$ was observed in the short-day strawberry cultivar “Toyonoka”, particularly reaching a difference of 0.035 under the 9-day treatment. Conversely, across all treatment temperatures and durations for “Toyonoka” as well as for the long-day strawberry cultivar “Selva”, the maximum differences from the actual photosynthetic efficiency remained largely within 0.05. An amplified difference between Fv/Fm and ${Fv}^{\prime }/{Fm}^{\prime }$ typically signifies limitations in the plant’s capacity to capture and utilize light energy [39]. However, we found a plausible explanation through the alterations in 1−qP and Y(NPQ) under corresponding chilling conditions. “Toyonoka” actively enhanced regulatory photoprotective mechanisms, simultaneously augmenting the active dissipation of excess energy while spontaneously restricting energy absorption [40]. This behavior aids in better confronting and mitigating the adverse effects brought about by chilling stress, thus preserving growth and survival in cold environments. 1−qP, an infrequently mentioned metric in most chlorophyll fluorescence studies, holds significant implications. It refers to the proportion of closed PSII reaction centers, indicating that under stress conditions, certain PSII reaction centers enter a closed state to shield PSII from damage. “Toyonoka” exhibited stable and controlled decreases in Y(Ⅱ) and sustained increases in Y(NO) under 15/5 °C and 10/0 °C treatments, progressively adjusting electron transport chain activity with escalating stress and enhancing passive energy dissipation. Conversely, “Selva” reached near-maximal levels of Y(NO) after 3 days under 10/0 °C, indicating the limited self-regulatory capability of the photosynthetic system in response to chilling stress.

SIF, operating as a passive indicator of chlorophyll fluorescence, possesses the advantage of not requiring dark adaptation or artificial light source excitation. This attribute renders it exceptionally well-suited to non-destructive and swift monitoring of changes in plant photosynthetic physiological performance within natural environments [41]. In recent years, SIF has been extensively applied in large-scale remote sensing inversions, encompassing assessments of plant drought stress severity [42], primary productivity [43] and vegetation phenology observations [44], and it has even found utility in leaf-scale plant disease diagnosis [45] and chlorophyll content estimation [46]. However, there remains a scarcity of research endeavors utilizing SIF as a diagnostic tool to evaluate the degree of plant response to low-temperature stress. This study highlights that both SIF and rSIF can accurately indicate the extent of damage in strawberries subjected to varying degrees of chilling stress. Notably, rSIF exhibits more stable changes in response to escalating stress compared to SIF, and its trend aligns more closely with $A_{max}$ and ${Fv}^{\prime }/{Fm}^{\prime }$. Furthermore, our investigation reveals that fluorescence intensity extracted from the O_2_-B atmospheric absorption band is numerically lower than that from the O_2_-A band. However, both extractions exhibit similar trends in SIF and rSIF. Notably, according to the PCA results’ loading plots, fluorescence values extracted from the O_2_-B band are more representative.

Upon evaluating the stress scores and grading outcomes of the photosynthetic systems in different photoperiodic strawberry varieties subjected to various degrees and durations of low-temperature stress, the CSPC emerges as a more robust indicator compared to individual photosynthetic physiological metrics. CSPC’s responsiveness to the fluctuations in chilling severity and stress duration exhibits greater stability, effectively circumventing the influence of inherent differences in physiological levels among distinct photoperiodic strawberry types. This approach distinctly portrays the adaptive recovery effect of “Toyonoka” under continuous treatment at 20/10 °C. These findings underscore the applicability of the study results across diverse photoperiodic strawberry varieties, and also reveal that under the same low-temperature environment, the photosynthetic system of long-day strawberry cultivars is more susceptible to stress-induced damage than their short-day counterparts.

In the study of hyperspectral inversion for assessing chilling injury severity on the strawberry photosynthetic system, spectral processing and wavelet transformation undoubtedly offer a wealth of distinctive spectral features. However, our observations reveal that the correlation between the wavelet coefficients of the high-frequency components obtained through wavelet analysis, after the removal of the spectral envelope, and the stress scores is significantly stronger than when solely applying smoothing filtration or first-order differentiation after a single-season filter. Notably, within the six sets of characteristic spectral wavelength bands that exhibit a correlation coefficient surpassing 0.5 with CSPC, the bands are concentrated within 576.46–614.83 nm, 826.23–866.15 nm, and 882.18–908.99 nm. These six feature spectral sets all carry physiological implications for the actual leaf. Chlorophyll absorption peaks were within the wavelength range of 576.46–614.83 nm, directly influencing the reflectance spectral intensity of this band [47]. The ranges of 826.23–866.15 nm and 882.18–908.99 nm belong to the near-infrared spectrum, with their reflectance spectral intensity correlated to leaf moisture content, stomata, and the compactness of leaf cellular structures including chloroplasts [48]. During the process of extracting spectral features associated with chilling stress, we attempted three classic spectral indices—ratio, normalized, and difference operations—aiming to enhance the correlation between feature band spectral intensities and CSPC. Among these operations, the difference operation displayed the most favorable enhancement effect. It is noteworthy that the prediction performance of the learning model based on RR significantly deviated from the actual CSPC values. With training utilizing 352 sets of spectral feature data, nearly none of the 80 predicted results approximated the actual values. Even if minimal alignment with the ranking predictions existed, we considered this to be purely coincidental. Both methods yielded results clustering around 0, which is the average value of all low-temperature stress scores. This underscores the considerable complexity of the mathematical relationships underlying the hyperspectral inversion of strawberry photosynthetic system’s chilling injury severity. Linear regression learning models fail to effectively extract the interconnections among data. The SVR algorithm, based on a nonlinear regression learning model, proved superior to RR, despite achieving only approximately 50% for fitting coefficients and accuracy. This indicates limited data mining performance in shallow regression models. In contrast, RF and BPNN, as the most popular machine learning and deep learning algorithms in recent years, demonstrated outstanding performance in this study, based on the original and wavelet hyperspectral feature indices for assessing the severity of chilling injury to the photosynthetic system in strawberries.

Based on the aforementioned theoretical research findings, we posit that for the future practical cultivation of strawberries, it is feasible to deploy hyperspectral measuring devices covering the wavelength ranges of 576.46–614.83 nm, 826.23–866.15 nm, and 882.18–908.99 nm above the plant canopy. By integrating machine learning algorithm models, this approach could facilitate the rapid, quantitative, non-contact monitoring of chilling injury to the photosynthetic system of strawberries. The findings presented in this study are drawn from observations made on the short-day strawberry cultivar “Toyonoka” and the long-day strawberry variety “Selva”. For additional strawberry varieties and under varying temperature differentials, whether the degree of chilling stress in plants continues to adhere to the patterns summarized in this study, and whether the model parameters can be widely and universally applicable, necessitate further experimentation. We need to continue conducting more low-temperature control experiments involving a broader range of strawberry varieties and diverse environmental variables in order to verify and refine these findings.

## 4. Materials and Methods

### 4.1. Plant Materials

In this study, two widely cultivated strawberry varieties in China, the short-day variety “Toyonoka” and the long-day variety “Selva”, were selected as the experimental plants. The experiments were conducted in two consecutive years, from November 2021 to January 2022 and from November 2022 to December 2022. The strawberry plants were obtained from a strawberry breeding base on 5 November 2021 and 9 November 2022, respectively. These plants were newly grown strawberry seedlings of the respective years and were not yet in the flowering stage at the time of purchase. Strawberry plants exhibiting 4–5 functional leaves, healthy foliage, sturdy main stems, and leaf stalks with an approximate height of 15 cm were transplanted into breathable grid-shaped pots (Figure 13). The pot height was 12 cm and the diameters of the top and the bottom were 15 cm and 11 cm, respectively. Each pot was filled with a substrate composed of vermiculite, perlite, peat moss, and garden soil in a 1:1:1:3 (*v*:*v*:*v*:*v*) ratio, totaling approximately1.2 kg per pot. Each pot was planted with a single strawberry plant. For each treatment, three replicates were adopted each year, resulting in a total of six replicates over the two-year period. In total, 72 plants of each variety were used, amounting to a combined usage of 144 plants for the two different photoperiod strawberry varieties.

### 4.2. Experimental Management and Treatment

Before conducting the chilling treatment experiments, all test plants were cultivated in a Venlo-type glass greenhouse at Nanjing University of Information Science and Technology (NUIST) with standardized water, fertilizer, and microclimate management. The grid-shaped pots containing strawberry plants were arranged within the ground planting beds. The gaps between the pots and the bed were filled with the same substrate, almost entirely embedding the pots within the bed. The dimensions of the beds were 1.2 m (length) × 3 m (width) × 0.2 m (depth). This approach facilitated efficient water and heat exchange between the substrate near the strawberry roots and in the bed, simulating the field planting scenario while allowing easy removal of plants for experimental treatments and physiological observations during the low-temperature control experiments. The substrate received initial fertilization with nitrogen (urea, 46% N, 150 kg/ha), phosphorus (calcium superphosphate, 12% P_2_O_5_, 200 kg/ha), and potassium (potassium sulfate, 52% K_2_O, 250 kg/ha). No additional fertilization was administered thereafter. Irrigation was conducted using the “5-point sampling method”, with water supplied to saturation when the average substrate moisture content decreased to 60%. The irrigation time ranged from 16:00 to 18:00. Throughout the greenhouse cultivation period, the microclimate conditions were controlled to maintain the optimal temperature and humidity range for strawberry growth (Figure 14). It is worth noting that the microclimate data were automatically recorded every 10 min, and the relative humidity exhibited fluctuations due to ventilation, irrigation, and other operations. The chilling treatment experiment was conducted when the strawberry plants had at least one fruit (fruit length ≥ 1.5 cm) and two flowers in bloom.

The low-temperature control experiments and physiological parameter observations were conducted at the NUIST Agricultural Meteorological Experimental Station (32°12′ N, 118°42′ E) from December 2021 to January 2022 and December 2022. The temperature control experiment included three stress levels: 20 °C (daily maximum temperature)/10 °C (daily minimum temperature), 15 °C/5 °C, and 10 °C/0 °C, with a control group (CK) set at 25 °C/15 °C. Three treatment durations of 3 days, 6 days, and 9 days were applied to each stress level. The information for each treatment is presented in Table 6.

Low-temperature control experiments were conducted inside a walk-in growth chamber (PGC-FLEX, Conviron, Canada), where the chamber’s temperature was dynamically adjusted to simulate the hourly fluctuations in greenhouse temperature trends (Figure 15) [49]. The chamber was subjected to a photoperiod of 12 h (7:00 to 19:00), with photosynthetically active radiation (PAR) maintained at 800 $\mu molm^{-2}s^{-1}$, and the relative humidity of the air was maintained between 60% and 70%. The time point for initiating and concluding the plant treatments was 8:00. Observations commenced at 9:00, with sequential measurements of photosynthetic parameters, chlorophyll fluorescence parameters, SIF, and leaf reflectance spectra. In each annual batch of experiments, there were 3 replicates for photosynthetic parameters, and 9 replicates for chlorophyll fluorescence and SIF parameters in each treatment. The number of replicates was doubled over the course of the two years.

### 4.3. The Methods of Measurement

#### 4.3.1. Photosynthetic Parameters

Between 9:00 and 12:00, we evaluated strawberry leaf photosynthetic parameters using a portable photosynthesis system (Li-6400xt, LI-COR, Lincoln, NE, USA). Our focus was directed towards the central leaflet of the second or third robust functional leaf, commencing from the uppermost point of the strawberry plants [50]. The instrument’s flow rate was configured to 500 $\mu mols^{-1}$, while the leaf chamber temperature was maintained at 25 °C. Relative humidity stood at 65%, and the reference chamber’s CO₂ concentration was upheld at 400 $\mu mol{mol}^{-1}$. The PAR for the analyzer was set to 18 gradients in the following order: 600, 1200, 1800, 1800, 1800, 1600, 1400, 1200, 1000, 800, 600, 400, 200, 150, 100, 50, 20, and 0 $\mu mol{m^{-2}s}^{-1}$. The instrument automatically recorded various photosynthetic parameters, including (A), stomatal conductance ($g_{s}$), transpiration rate (E), intercellular ${CO}_{2}$ concentration (ci), and atmospheric ${CO}_{2}$ concentration (ca), under different simulated light intensities according to the set program. Water-use efficiency (WUE) was calculated using Equation (1) [51]. A light response model was employed to fit the maximum net photosynthetic rate ($A_{max}$), light compensation point (LCP), light saturation point (LSP), and apparent quantum efficiency (AQE) [52,53]. The values shown in this study represent the average values of plants in a stable state.
(1)WUE=A/E

#### 4.3.2. Chlorophyll Fluorescence Induction Kinetics Parameters

Using a portable modulated chlorophyll fluorometer (MINI-PAM, Walz, Germany), we conducted measurements of chlorophyll fluorescence induction kinetics on strawberry leaves from 9:00 to 11:00 [54]. Three healthy, central leaflets were selected as observation targets from the second to fifth functional leaves of the plant, proceeding from the top. These leaves were consistent with the subsequent measurements of SIF and reflectance spectra. Prior to the initial measurements, the instrument’s “Gain” was adjusted to set the fluorescence value (Ft) between 200 and 400 when only the measuring light was on. Before each measurement, the leaves were dark-adapted for 30 min by clipping them with leaf clips, and measurement points were chosen to avoid major veins. During the measurements, at each point, two saturating pulses were applied consecutively. Upon the first application of the saturating pulse, the instrument automatically recorded $F_{0}$, $F_{m}$, and Y(Ⅱ). After a brief waiting period until Ft stabilized, the saturating pulse was applied again, and the instrument automatically recorded $F_{0}^{\prime }$, $F_{m}^{\prime }$, qP, qN, qL, Y(NPQ), and Y(NO).

#### 4.3.3. Solar-Induced Chlorophyll Fluorescence

In the clear open-air conditions, we utilized a fiber optic hyperspectral spectrometer (QE65 Pro, Ocean Optics, Orlando, FL, USA) to collect leaf reflectance radiance spectra and obtained the “whiteboard” reflectance radiance as the incident solar irradiance spectrum. The observation was conducted from 11:00 to 12:00. The instrument’s operating spectral range was 645–800 nm, with a spectral resolution of 1.4 nm. During the observations, the leaves and the “whiteboard” were placed horizontally parallel to the ground, and a fixed setup maintained the leaves perpendicular to the fiber optic probe. Each measurement was repeated twice. In this study, the solar-induced chlorophyll fluorescence (SIF) was extracted using the radiance-based 3FLD algorithm [55]. This method assumes a linear variation of reflectance and chlorophyll fluorescence spectra across atmospheric absorption lines. It employs the weighted average of radiance from one spectral band on each side of the absorption line to fit the reflectance radiance and solar irradiance in the absorption line’s central band. Based on these three bands, the chlorophyll fluorescence intensity within the absorption line is calculated. The specific calculation method is given by Equations (2)–(4).
(2)$$\omega_{left}=\frac{\lambda_{right}-\lambda_{in}}{\lambda_{right}-\lambda_{left}}$$
(3)$$\omega_{right}=\frac{\lambda_{in}-\lambda_{left}}{\lambda_{right}-\lambda_{left}}$$
(4)$$F_{in}=\frac{L_{in}*\left(\omega_{left}*I_{left}+\omega_{right}*I_{right}\right)-\left(\omega_{left}*L_{left}+\omega_{right}*L_{right}\right)*I_{in}}{\left(\omega_{left}*I_{left}+\omega_{right}*I_{right}\right)-I_{in}}$$

In the equation, $\omega_{left}$ and $\omega_{right}$ represent the weights of the reference bands on the left and right sides of the atmospheric absorption line, respectively; $\lambda_{in}$, $\lambda_{left}$, and $\lambda_{right}$ denote the wavelengths within the absorption line, on the left side, and on the right side, respectively; $F_{in}$ corresponds to the solar-induced chlorophyll fluorescence intensity within the absorption line; $L_{in}$, $L_{left}$, and $L_{right}$ represent the radiance spectra intensity of strawberry leaf reflectance within the absorption line, on the left side, and on the right side, respectively; $I_{in}$, $I_{left}$, and $I_{right}$ refer to the solar irradiance spectra intensity within the absorption line, on the left side, and on the right side, respectively.

#### 4.3.4. Reflectance Spectra

We employed a portable hyperspectral imaging system (SOC710, Surface Optics Corporation, San Diego, CA, USA) to capture the reflectance spectra of strawberry leaves. The observations were conducted indoors from 12:00 to 14:00, using two 75 W tungsten halogen lamps (PAR30, OSRAM, Munich, Germany) as artificial light sources. The instrument’s spectral range was 375.88–1039.19 nm, and we selected the 256-band mode with a spectral resolution of 2.6 nm for this study. The instrument was positioned vertically over the observation target, and after performing reference panel reflectance calibration on the acquired data, 2 or 3 regions of interest (ROI) were selected within the observation area of each target leaf. The average spectra of ROIs were used as the reflectance spectra of the target leaves.

### 4.4. Data Processing and Analysis

#### 4.4.1. Hyperspectral Data Preprocessing

Due to the narrow band range of hyperspectral data, the signal-to-noise ratio is lower and data stability is weaker compared to spectral data with broader band ranges. This results in apparent sawtooth patterns and severe random fluctuations near the starting and maximum wavelengths on the spectral curve. To address these challenges, we selected a relatively stable range of 410.49–990.07 nm from the original spectral data for our study. Furthermore, a non-linear mean filter was applied to reduce noise interference [56]. After conducting repeated tests, we found that using a sliding convolution window size of 13 and fitting a polynomial of degree 3 yielded satisfactory filtering results. This approach effectively preserved the spectral details while suppressing noise. Subsequently, two types of processing were performed on the filtered spectral data: envelope removal and first-order differentiation. This provided us with three distinct sets of spectral data representing different characteristics. The spectral preprocessing was implemented using MATLAB R2021b programming language.

#### 4.4.2. Statistical and Analytical Methods

For statistical analyses in this investigation, we employed IBM SPSS Statistics version 24 (IBM Corp., Chicago, IL, USA). One-way analysis of variance (ANOVA), accompanied by Duncan’s multiple comparison test (*p* = 0.05) and principal component analysis (PCA), were conducted. All observed parameters were reported as “Mean ± SD” (mean ± standard deviation).

PCA, widely employed for data dimensionality reduction, was utilized in this research to extract the main features of multiple data by focusing on the high contribution rates of selected indicators [57]. This method was applied for the feature extraction of different types of photosynthetic physiological parameters and the construction of a comprehensive evaluation index for chilling stress. The computation methods for weights of each principal component and the comprehensive scores are depicted in Equations (5)–(8) [58].
(5)$$W_{ki}=\frac{L_{ki}}{\sqrt{E_{k}}}$$
(6)$${CS}_{k}=\sum_{i-1}^{n}W_{ki}\times X_{i}$$
(7)$$W_{k}=\frac{P_{k}}{\sum_{k-1}^{n}P_{k}}$$
(8)$$CS=\sum_{k-1}^{n}W_{k}\times {CS}_{k}$$

In the equation, $W_{ki}$ represents the weight of the i^th^ indicator in principal component A_k_ (PC_Ak_) (k = 1, 2, …, *p*); $L_{ki}$ denotes the loadings of the i^th^ indicator of A_k_; $E_{k}$ corresponds to the eigenvalues of A_k_; ${CS}_{k}$ represents the composite score of A_k_; $X_{i}$ refers to the normalized value of the i^th^ indicator (i = 1, 2, …, n); $W_{k}$ denotes the weight of PC_Ak_; $P_{k}$ corresponds to the contribution rate of PC_Ak_; CS represents the final PCA composite score.

The coefficient of determination (R^2^), root mean square error (RMSE), and accuracy (Acc) were employed to quantify the fitting accuracy and predictive performance of the model. The calculation formulas are presented in Equations (9) and (11).
(9)R2=∑in(yi^−y¯)2∑in(yi−y¯)2
(10)RMSE=∑in(yi^−yi)2n
(11)Acc=ntrue^n

In the equation, yi^ represents the predicted value; $y_{i}$ represents the actual value; $\bar{y}$ denotes the mean value; n corresponds to the number of validation samples; ntrue^ denotes the count of validation samples that were accurately predicted.

#### 4.4.3. Wavelet Transformation

Wavelet transformation is an important signal processing method used to analyze digital data [59]. It decomposes the signal into one low-frequency wavelet signal and the same number of high-frequency wavelet signals as the transformation levels. The low-frequency signal reflects the overall characteristics of the spectrum, while the high-frequency signals capture its detailed features [60]. To capture spectral details that are highly correlated with the comprehensive score of the photosynthetic system under chilling stress, this study employed the Bior1.3 biorthogonal wavelet function to perform wavelet transformation on the filtered, envelope-removed, and first-order differentiated spectra, in 1–6 decomposition levels. This allowed the extraction of low-frequency wavelet coefficient spectra corresponding to scales 2^0^, 2^1^, 2^2^, 2^3^, 2^4^, and 2^5^. Thus, a comprehensive exploration of the hyperspectral information was achieved. Wavelet transformation was implemented using the “wavedec” function in MATLAB R2021b, following the general transformation formulas as depicted in Equations (12) and (13) [61].
(12)$$\psi_{a,b}(\lambda )=\frac{1}{\sqrt{a}}\times \psi (\frac{\lambda -b}{a})$$
(13)$$W_{f}\left(a,b\right)=<f\left(\lambda \right),\psi_{a,b}\left(\lambda \right)>=\int_{-\infty }^{+\infty }f\left(\lambda \right){\times \psi }_{a,b}\left(\lambda \right)d\lambda$$

In the equation, $\psi_{a,b}(\lambda )$ represents the wavelet mother function; λ denotes the spectral band index; a and b denote the scale and phase of the wavelet, respectively; $W_{f}\left(a,b\right)$ corresponds to the wavelet coefficient matrix; and $f\left(\lambda \right)$ refers to the spectral reflectance before transformation at the $\lambda^{th}$ spectral band.

#### 4.4.4. Hyperspectral Inversion Modeling

We established hyperspectral inversion models for the comprehensive score of the strawberry photosynthetic system under chilling stress using four different machine learning algorithms: Backpropagation Neural Network (BPNN) [62], Random Forest (RF) [63], Support Vector Regression (SVR) [64], and Ridge Regression (RR) [65]. The modeling performance of these distinct mechanisms in hyperspectral modeling was compared. The BPNN employed in this study comprised 8 hidden layers: 2 with 12 neurons, 2 with 24 neurons, 2 with 48 neurons, and 2 with 6 neurons. The Rectified Linear Unit (ReLU) activation function was used, and Dropout layers were included before the fully connected layer to prevent overfitting. The model was trained for 1500 iterations. The RF was configured with 100 trees, SVR used the radial basis function (RBF) with a gamma coefficient of 0.1, and RR was set with an alpha coefficient of 1.0. All inversion models were developed using Python 3.8 programming language.

## 5. Conclusions

In experiments conducted at three dynamic chilling conditions of 20/10 °C, 15/5 °C, and 10/0 °C, with stress durations of 3, 6, and 9 days, the cold tolerance of short-day strawberry cultivar “Toyonoka” surpassed that of the long-day representative “Selva”. Under continuous treatment at 20/10 °C, after 6 days, different photoperiod strawberry types exhibited initial signs of cold-induced physiological stress and escalating stress levels evident by the 9th day, with “Selva” experiencing aggravated stress while “Toyonoka” rebounded to an unstressed state. At 15/5 °C and 10/0 °C conditions, the stress imposed on “Selva” surpassed that on “Toyonoka” with equivalent treatment durations. The short-day “Toyonoka” encountered stress of intensity greater than or equal to three levels after 9 days at 15/5 °C and 6 days at 10/0 °C, while the long-day “Selva” experienced stress of no less than three levels after 6 days at 15/5 °C and 3 days at 10/0 °C. The critical temperature for chilling injury in the photosynthetic system of short-day strawberry was lower than that of the long-day strawberry.

WUE, LCP, Y(Ⅱ), qN, ${SIF}_{O_{2}-B}$, and ${rSIF}_{O_{2}-B}$ were identified as feature indicators reflecting the overall characteristics of photosynthetic parameters, chlorophyll fluorescence induction kinetics parameters, and solar-induced chlorophyll fluorescence parameters in strawberry plants. The Comprehensive Score for Photosynthetic System Chilling Injury (CSPC) established from these six photosynthetic fluorescence features quantitatively captured the variations in photosynthetic system physiological stress severity and low-temperature adaptation capabilities across different photoperiod strawberry types subjected to varying degrees of low temperature and duration. The first five layers of high-frequency components obtained through hyperspectral continuous wavelet transformation effectively unearthed feature band information closely correlated with the degree of cold-induced stress in the strawberry photosynthetic system, offering optimal spectral features for hyperspectral inversion modeling. The correlation coefficient of the mean-filtered spectral CD1, located at 882.18 nm in the near-infrared band, reached the highest value at 0.534. Among the four models compared for the hyperspectral inversion modeling of strawberry photosynthetic system cold stress, the comprehensive performance ranked from highest to lowest as BPNN, RF, SVR, and RR. If prediction of the chilling injury severity score is required in practical production, the RF algorithm is recommended. For predicting only the stress level, BPNN emerges as a superior choice. Conversely, the linear regression learning model based on the RR algorithm failed to establish a significant correlation between spectral features and the cold stress rating index, rendering it unsuitable for the hyperspectral inversion modeling employed in this study. This research offers scientific guidance for selecting suitable strawberry cultivars and addressing chilling stress in the photosynthetic system during flowering and fruiting. It also provides non-destructive diagnostic solutions for practical production.

In future endeavors, we envision expanding our investigations to encompass strawberry cultivars belonging to diverse photoperiod types. This expansion aims to unravel the physiological significance and inherent interconnections among distinctive spectral signatures. We also intend to delve into the ramifications of low-temperature stress during various developmental stages on strawberry yield and fruit quality, and continually enhance our understanding of the physiological and production-related impacts of cold stress on strawberries.

## Figures and Tables

**Figure 1 plants-12-03138-f001:**
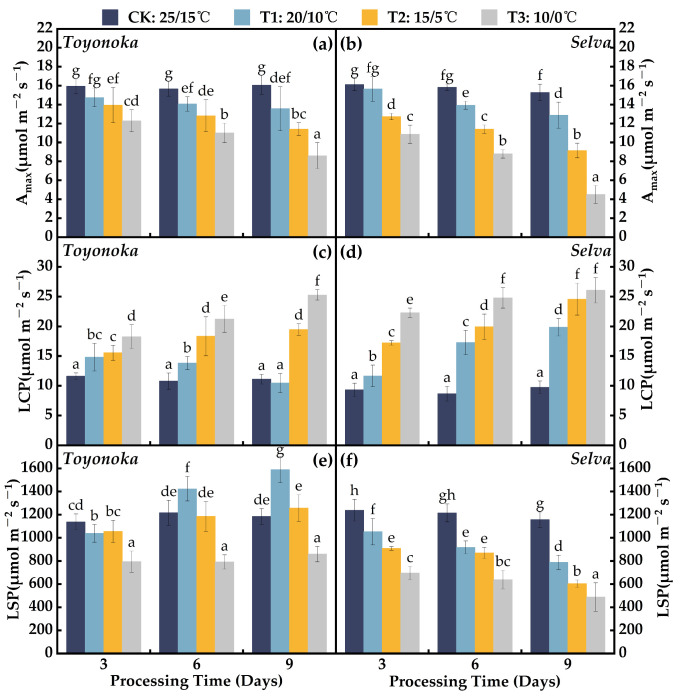
Changes in light response parameters of short-day and long-day strawberry varieties under continuous chilling: Panels (**a**,**c**,**e**) represent the maximum net photosynthetic rate (A_max_), light compensation point (LCP), and light saturation point (LSP) of the short-day strawberry cultivar “Toyonoka”, respectively. Panels (**b**,**d**,**f**) correspondingly illustrate the A_max_, LCP, and LSP of the long-day cultivar “Selva”. Each value is presented as “mean ± standard deviation (SD)” in the figure. Distinct lowercase letters are used to denote significant differences among treatments at the *p* < 0.05 level, as determined by Duncan’s test. The temperature settings of 25/15 °C, 20/10 °C, 15/5 °C, and 10/0 °C are denoted as CK, T1, T2, and T3, respectively. The durations of 3 days, 6 days, and 9 days are indicated by D3, D6, and D9, respectively.

**Figure 2 plants-12-03138-f002:**
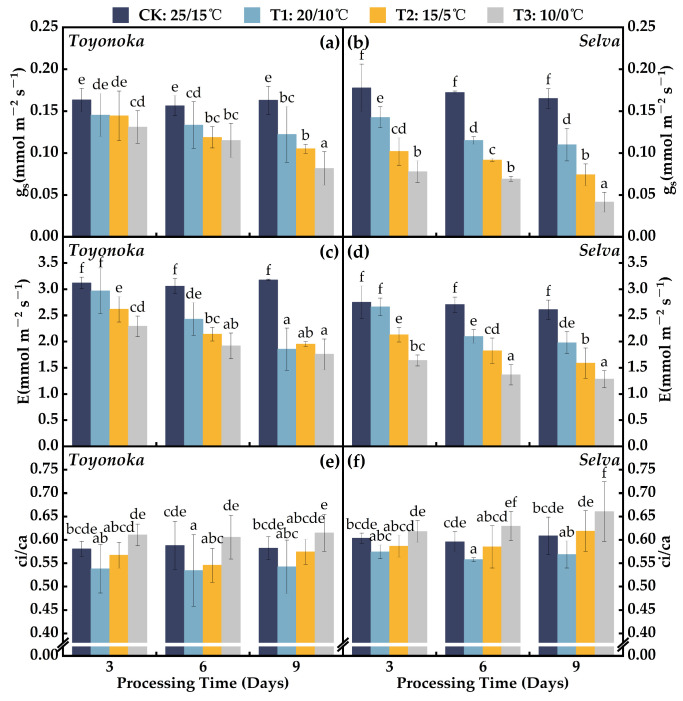
Changes in gas exchange parameters of short-day and long-day strawberry varieties under continuous chilling: Panels (**a**,**c**,**e**) represent the stomatal conductance ($g_{s}$), transpiration rate (E), and the ratio of intercellular CO_2_ concentration to atmospheric CO_2_ concentration (ci/ca) of the short-day strawberry cultivar “Toyonoka”, respectively. Panels (**b**,**d**,**f**) correspondingly illustrate the $g_{s}$, E, and ci/ca of the long-day cultivar “Selva”. Each value is presented as “mean ± standard deviation (SD)” in the figure. Distinct lowercase letters are used to denote significant differences among treatments at the *p* < 0.05 level, as determined by Duncan’s test. The temperature setting $g_{s}$ values of 25/15 °C, 20/10 °C, 15/5 °C, and 10/0 °C are denoted as CK, T1, T2, and T3, respectively. The durations of 3 days, 6 days, and 9 days are indicated by D3, D6, and D9, respectively.

**Figure 3 plants-12-03138-f003:**
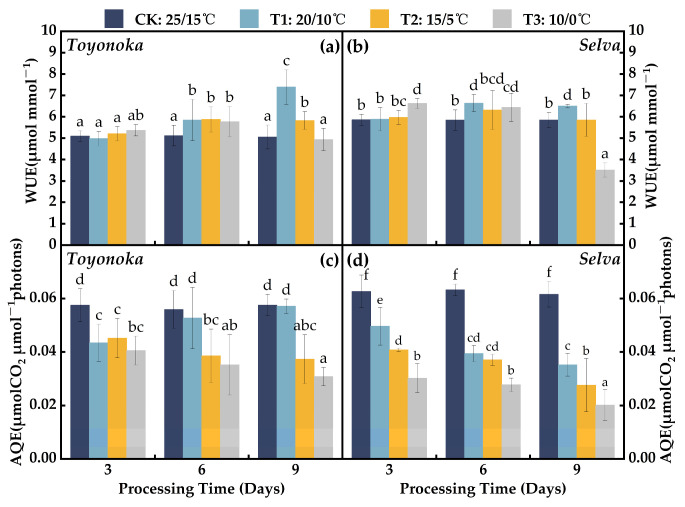
Changes in parameters of water and light use efficiency of short-day and long-day strawberry varieties under continuous chilling: Panels (**a**,**c**) represent the water use efficiency (WUE) and apparent quantum efficiency (AQE) of the short-day strawberry cultivar “Toyonoka”, respectively. Panels (**b**,**d**) correspondingly illustrate the WUE and AQE of the long-day cultivar “Selva”. Each value is presented as “mean ± standard deviation (SD)” in the figure. Distinct lowercase letters are used to denote significant differences among treatments at the *p* < 0.05 level, as determined by Duncan’s test. The temperature settings of 25/15 °C, 20/10 °C, 15/5 °C, and 10/0 °C are denoted as CK, T1, T2, and T3, respectively. The durations of 3 days, 6 days, and 9 days are indicated by D3, D6, and D9, respectively.

**Figure 4 plants-12-03138-f004:**
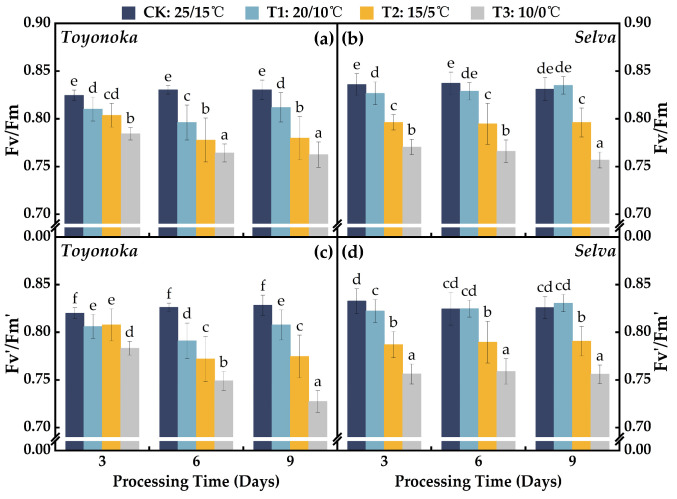
Changes in photosystem II photochemical efficiency parameters of short-day and long-day strawberry varieties under continuous chilling: Panels (**a**,**c**) represent the maximum PSII photochemical efficiency (Fv/Fm) and actual PSII photochemical efficiency (${Fv}^{\prime }/{Fm}^{\prime }$) of the short-day strawberry cultivar “Toyonoka”, respectively. Panels (**b**,**d**) correspondingly illustrate the Fv/Fm and ${Fv}^{\prime }/{Fm}^{\prime }$ of the long-day cultivar “Selva”. Each value is presented as “mean ± standard deviation (SD)” in the figure. Distinct lowercase letters are used to denote significant differences among treatments at the *p* < 0.05 level, as determined by Duncan’s test. The temperature settings of 25/15 °C, 20/10 °C, 15/5 °C, and 10/0 °C are denoted as CK, T1, T2, and T3, respectively. The durations of 3 days, 6 days, and 9 days are indicated by D3, D6, and D9, respectively.

**Figure 5 plants-12-03138-f005:**
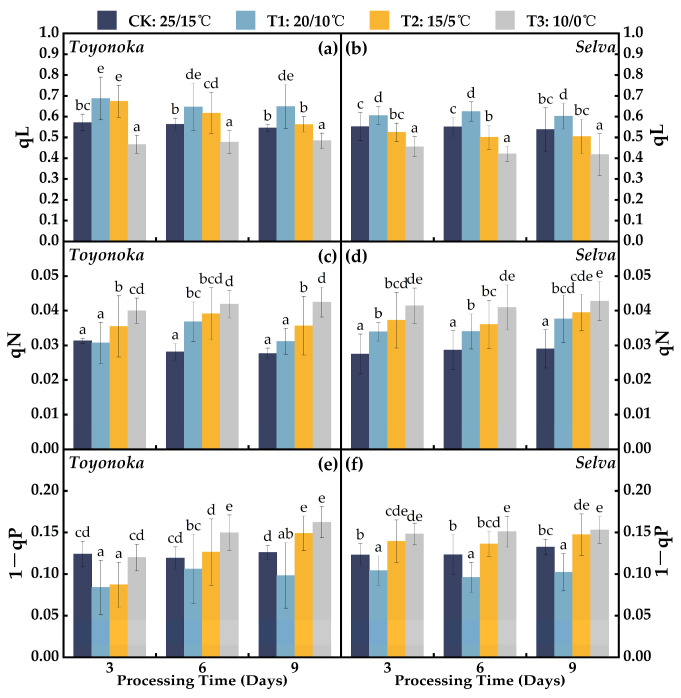
Changes in fluorescence quenching coefficients of short-day and long-day strawberry varieties under continuous chilling: Panels (**a**,**c**,**e**) represent the lake-type non-photochemical quenching coefficients (qL), the non-photochemical quenching coefficient (qN), and the redox state of Q_A_ (1−qP) of the short-day strawberry cultivar “Toyonoka”, respectively. Panels (**b**,**d**,**f**) correspondingly illustrate the qL qN, and 1−qP of the long-day cultivar “Selva”. Each value is presented as “mean ± standard deviation (SD)” in the figure. Distinct lowercase letters are used to denote significant differences among treatments at the *p* < 0.05 level, as determined by Duncan’s test. The temperature settings of 25/15 °C, 20/10 °C, 15/5 °C, and 10/0 °C are denoted as CK, T1, T2, and T3, respectively. The durations of 3 days, 6 days, and 9 days are indicated by D3, D6, and D9, respectively.

**Figure 6 plants-12-03138-f006:**
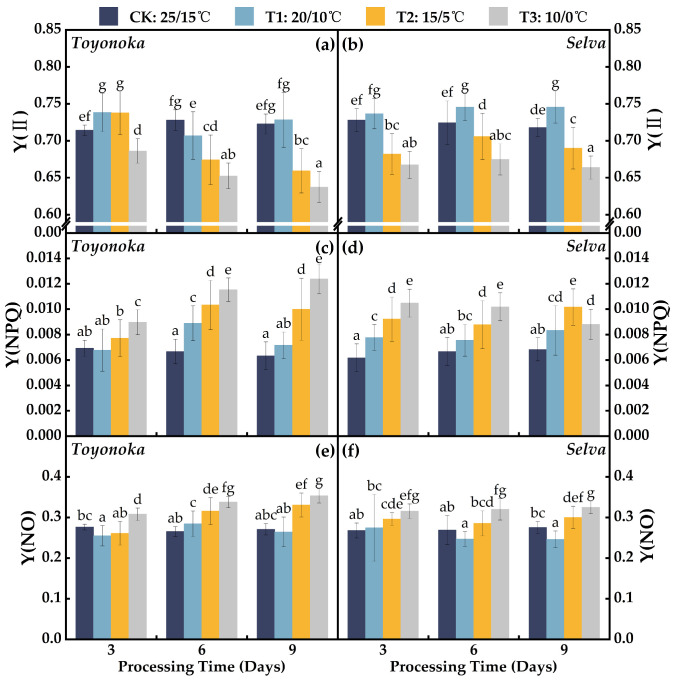
Changes in photosystem II quantum yield parameters of short-day and long-day strawberry varieties under continuous chilling: Panels (**a**,**c**,**e**) represent the PSⅡ quantum yield (Y(Ⅱ)), the PSII regulatory energy dissipation quantum yield (Y(NPQ)), and the PSII non-regulatory energy dissipation quantum yield (Y(NO)) of the short-day strawberry cultivar “Toyonoka”, respectively. Panels (**b**,**d**,**f**) correspondingly illustrate the Y(Ⅱ) Y(NPQ), and Y(NO) of the long-day cultivar “Selva”. Each value is presented as “mean ± standard deviation (SD)” in the figure. Distinct lowercase letters are used to denote significant differences among treatments at the *p* < 0.05 level, as determined by Duncan’s test. The temperature settings of 25/15 °C, 20/10 °C, 15/5 °C, and 10/0 °C are denoted as CK, T1, T2, and T3, respectively. The durations of 3 days, 6 days, and 9 days are indicated by D3, D6, and D9, respectively.

**Figure 7 plants-12-03138-f007:**
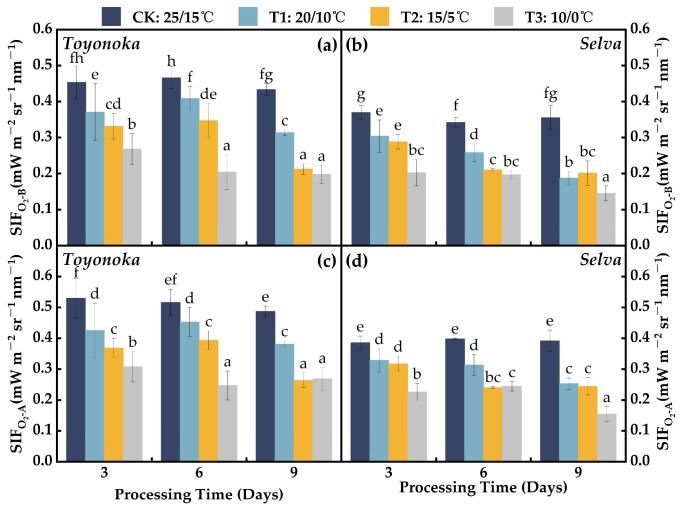
Changes in solar-induced chlorophyll fluorescence of short-day and long-day strawberry varieties under continuous chilling: Panels (**a**,**c**) represent the SIF retrieved based on the O_2_-B absorption band (${SIF}_{O_{2}-B}$) and O_2_-A absorption band (${SIF}_{O_{2}-A}$) of the short-day strawberry cultivar “Toyonoka”, respectively. Panels (**b**,**d**) correspondingly illustrate the ${SIF}_{O_{2}-B}$ and ${SIF}_{O_{2}-A}$ of the long-day cultivar “Selva”. Each value is presented as “mean ± standard deviation (SD)” in the figure. Distinct lowercase letters are used to denote significant differences among treatments at the *p* < 0.05 level, as determined by Duncan’s test. The temperature settings of 25/15 °C, 20/10 °C, 15/5 °C, and 10/0 °C are denoted as CK, T1, T2, and T3, respectively. The durations of 3 days, 6 days, and 9 days are indicated by D3, D6, and D9, respectively.

**Figure 8 plants-12-03138-f008:**
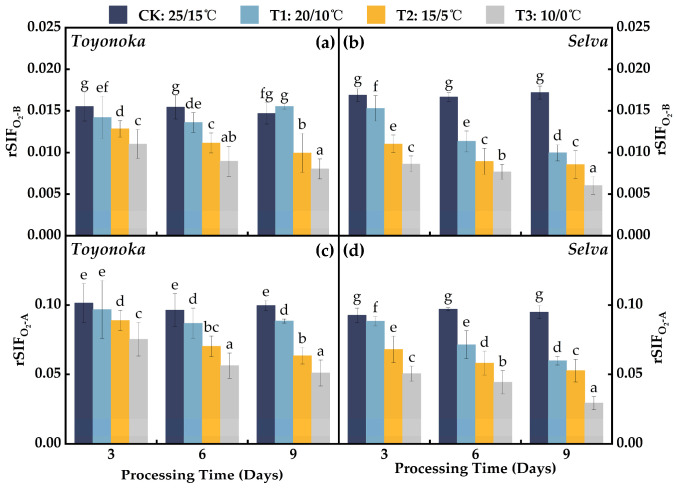
Changes in relative solar-induced chlorophyll fluorescence of short-day and long-day strawberry varieties under continuous chilling: Panels (**a**,**c**) represent the relative SIF retrieved based on the O_2_-B absorption band ($r{SIF}_{O_{2}-B}$) and O_2_-A absorption band (${rSIF}_{O_{2}-A}$) of the short-day strawberry cultivar “Toyonoka”, respectively. Panels (**b**,**d**) correspondingly illustrate the ${rSIF}_{O_{2}-B}$ and ${rSIF}_{O_{2}-A}$ of the long-day cultivar “Selva”. Each value is presented as “mean ± standard deviation (SD)” in the figure. Distinct lowercase letters are used to denote significant differences among treatments at the *p* < 0.05 level, as determined by Duncan’s test. The temperature settings of 25/15 °C, 20/10 °C, 15/5 °C, and 10/0 °C are denoted as CK, T1, T2, and T3, respectively. The durations of 3 days, 6 days, and 9 days are indicated by D3, D6, and D9, respectively.

**Figure 9 plants-12-03138-f009:**
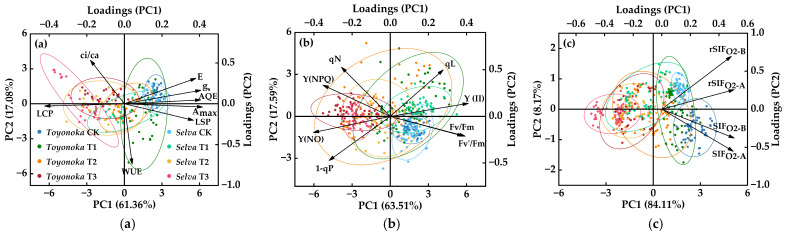
Principal component analysis scores and loadings of photosynthetic physiological indices in different photoperiod types of strawberries under prolonged dynamic chilling: Panels (**a**–**c**) present the PCA results for photosynthetic parameters, chlorophyll fluorescence induction dynamics parameters, and solar-induced chlorophyll fluorescence parameters, respectively. The PCA scores of the same temperature treatment group share the same color as the corresponding 95% confidence ellipses.

**Figure 10 plants-12-03138-f010:**
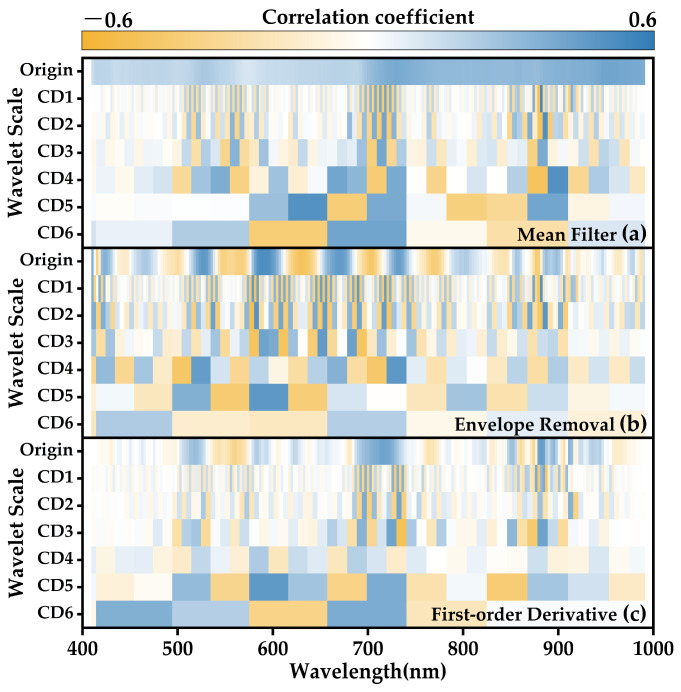
Correlation coefficients between original spectra and wavelet coefficients spectra at different scales with comprehensive scores for photosynthetic system chilling injury in strawberries: Panels (**a**–**c**) respectively display the correlation coefficients of mean filtering, envelope removal, and first-order derivative transformed original spectra, as well as the six-layer wavelet high-frequency components (CD1–CD6), with Comprehensive Score for Photosynthetic System Chilling Injury (CSPC).

**Figure 11 plants-12-03138-f011:**
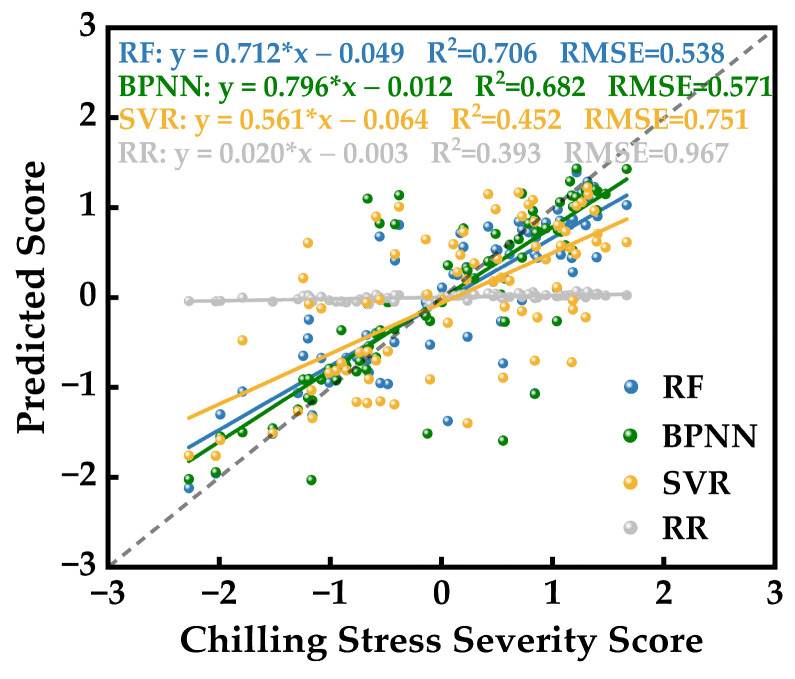
Validation of hyperspectral inversion for predicting comprehensive score for photosynthetic system chilling injury in strawberries. RF: Random Forest; BPNN: Backpropagation Neural Network; SVR: Support Vector Regression; RR: Ridge Regression.

**Figure 12 plants-12-03138-f012:**
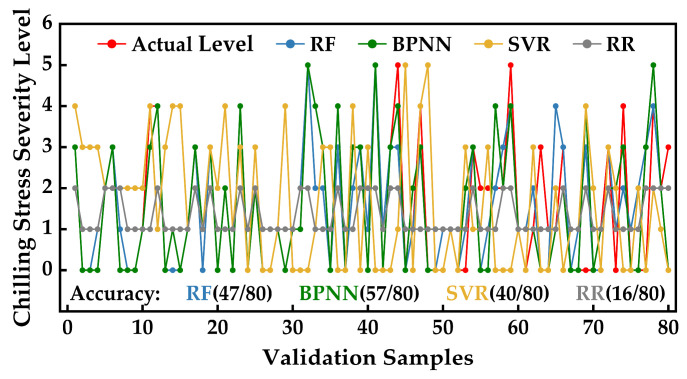
Validation of hyperspectral inversion for predicting stress levels for photosynthetic system chilling injury in strawberries. RF: Random Forest; BPNN: Backpropagation Neural Network; SVR: Support Vector Regression; RR: Ridge Regression. Accuracy is expressed as “number of correctly predicted samples/total number of validation samples”.

**Figure 13 plants-12-03138-f013:**
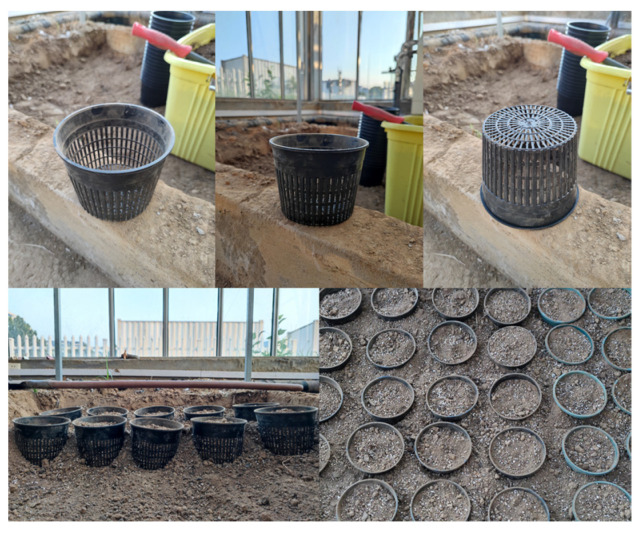
Illustration of cultivation using breathable grid-shaped pots and planting beds.

**Figure 14 plants-12-03138-f014:**
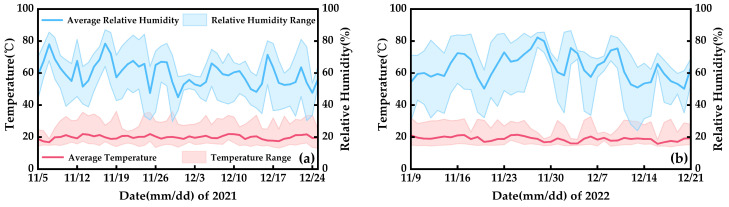
Historical temperature and relative humidity records during greenhouse cultivation in 2021 and 2022: Panels (**a**,**b**) represent the meteorological data records for the years 2021 and 2022, respectively.

**Figure 15 plants-12-03138-f015:**
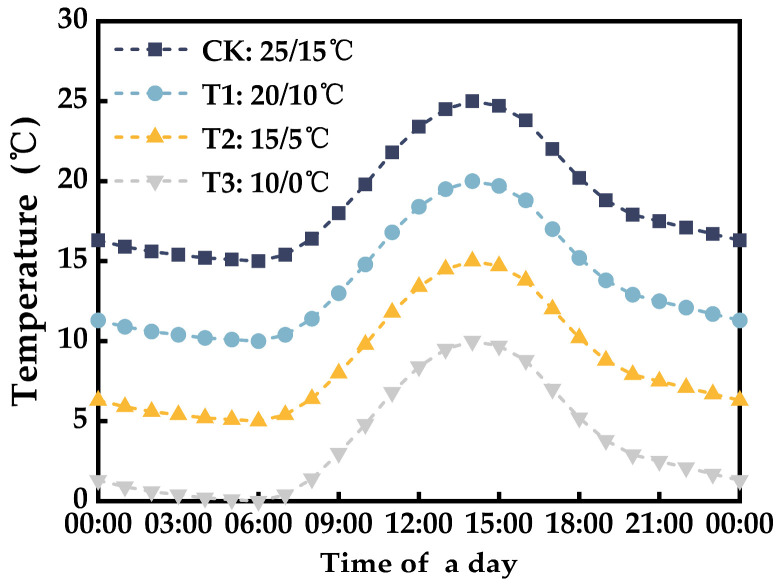
Hourly temperature settings in artificial climate chambers for different chilling treatment groups.

**Table 1 plants-12-03138-t001:** Variance contribution and loadings of characteristic indices in principal components analysis.

Principal Components		PC1	PC2	PC3
Eigenvalue		3.43	1.072	0.621
Variance contribution (%)		57.167	17.874	10.356
Cumulative variance contribution (%)		57.167	75.041	85.397
Loadings	WUE	0.047	0.926	−0.278
	LCP	−0.506	0.012	0.043
	Y(II)	0.359	0.281	0.800
	qN	−0.396	0.047	0.526
	${SIF}_{O_{2}-B}$	0.456	−0.247	−0.024
	$r{SIF}_{O_{2}-B}$	0.498	−0.013	−0.067

Note: WUE represents the water use efficiency; LCP denotes the light compensation point; Y(II) corresponds to the PSⅡ quantum yield; qN refers to the non-photochemical quenching coefficient; ${SIF}_{O_{2}-B}$ and $r{SIF}_{O_{2}-B}$ represent the SIF and rSIF retrieved based on the O_2_-B absorption band, respectively; PC1, PC2, and PC3 represent the top three principal components with the highest variance contribution.

**Table 2 plants-12-03138-t002:** Calculation formula for principal component scores of characteristic indices and comprehensive score for photosynthetic system chilling injury in strawberries.

Variables	Formulas
PC1 Score	$S_{1}=0.047*WUE-0.506*LCP+0.359*Y(Ⅱ)-0.396*qN+0.456*{SIF}_{O_{2}-B}+0.498*r{SIF}_{O_{2}-B}$
PC2 Score	$S_{2}=0.926*WUE+0.012*LCP+0.281*Y(Ⅱ)+0.047*qN-0.247*{SIF}_{O_{2}-B}-0.013*r{SIF}_{O_{2}-B}$
PC3 Score	$S_{3}=-0.278*WUE+0.043*LCP+0.800*Y(Ⅱ)+0.526*qN-0.024*{SIF}_{O_{2}-B}-0.067*r{SIF}_{O_{2}-B}$
Composite Score	$CSPC=0.572*S_{1}+0.179*S_{2}+0.104*S_{3}$

Note: The values of WUE, LCP, Y(Ⅱ), qN, ${SIF}_{O_{2}-B}$ and $r{SIF}_{O_{2}-B}$ in the above formulas are all standardized; $S_{1}$, $S_{2}$, and $S_{3}$ represent the principal component analysis scores for PC1, PC2, and PC3, respectively; CSPC denotes the Comprehensive Score for Photosynthetic System Chilling Injury.

**Table 3 plants-12-03138-t003:** Grading the severity of chilling injury on the photosynthetic system of strawberries.

Comprehensive Score for Photosynthetic System Chilling Injury	Strawberry Chilling Stress Severity Level
0.54 ≤ CSPC	Level 0
0 ≤ CSPC < 0.54	Level 1
−0.54 ≤ CSPC < 0	Level 2
−1.08 ≤ CSPC < −0.54	Level 3
−1.62 ≤ CSPC < −1.08	Level 4
CSPC < −1.62	Level 5

Note: CSPC—Comprehensive Score for Photosynthetic System Chilling Injury. A higher stress level indicates a more severe chilling stress on the strawberry’s photosynthetic system.

**Table 4 plants-12-03138-t004:** Chilling injury severity on the photosynthetic system of strawberries with different photoperiod types at various degrees and durations of low temperature.

Strawberry Chilling Injury Severity							
Varieties	Days of Duration	3 d		6 d		9 d	
	Daily Maximum Temperature/Daily Minimum Temperature	Score	Level	Score	Level	Score	Level
Toyonoka	25 °C/15 °C	0.947	0	1.202	0	1.009	0
	20 °C/10 °C	0.704	0	−0.250	1	1.234	0
	15 °C/5 °C	0.403	1	−0.339	2	−0.842	3
	10 °C/0 °C	−0.525	2	−1.231	4	−1.813	5
Selva	25 °C/15 °C	1.323	0	1.218	0	1.172	0
	20 °C/10 °C	0.876	0	0.397	1	−0.094	2
	15 °C/5 °C	−0.294	2	−0.470	3	−1.045	4
	10 °C/0 °C	−1.024	3	−1.593	4	−2.157	5

Note: Score—Comprehensive Score for Photosynthetic System Chilling Injury (CSPC); Level—severity level of strawberry chilling stress categorized according to CSPC division.

**Table 5 plants-12-03138-t005:** Correlation coefficients between typical spectral features and operations from spectral features with comprehensive scores for photosynthetic system chilling injury in strawberries.

Characteristic Spectral Information		Mean Filter		Envelope Removal		First-Order Derivative	
		Wavelength	Correlation Coefficient	Wavelength	Correlation Coefficient	Wavelength	Correlation Coefficient
Origin	Ba	954.82	0.417	589.22	0.516	882.18	0.492
	Bb	573.91	0.115	630.25	−0.480	558.64	−0.333
	Ba − Bb		0.440		0.521		0.472
	Ba/Bb		0.264		−0.314		−0.083
	(Ba − Bb)/(Ba + Bb)		0.273		0.052		0.060
CD1	Ba	882.18	0.534	882.18	0.495	884.86	0.391
	Bb	879.51	−0.508	581.56	−0.468	887.54	−0.417
	Ba − Bb		0.527		0.532		0.409
	Ba/Bb		−0.030		−0.053		−0.016
	(Ba − Bb)/(Ba + Bb)		0.032		−0.059		−0.071
CD2	Ba	884.86–887.54	0.510	581.56–584.12	0.492	731.40–734.02	0.411
	Bb	879.51–882.18	−0.481	576.46–579.01	−0.498	700.11–702.71	−0.376
	Ba − Bb		0.499		0.497		0.436
	Ba/Bb		−0.069		0.010		−0.187
	(Ba − Bb)/(Ba + Bb)		0.108		0.051		−0.009
CD3	Ba	710.52–718.34	0.398	648.29–656.03	0.451	720.95–728.79	0.424
	Bb	700.11–707.92	−0.378	576.46–584.12	−0.474	731.40–739.25	−0.485
	Ba − Bb		0.391		0.485		0.462
	Ba/Bb		0.277		0.046		−0.213
	(Ba − Bb)/(Ba + Bb)		−0.011		0.040		−0.092
CD4	Ba	890.21–908.99	0.504	720.95–739.25	0.469	576.46–594.34	0.237
	Bb	868.82–887.53	−0.473	495.40–513.04	−0.445	596.90–614.83	−0.187
	Ba − Bb		0.497		0.499		0.217
	Ba/Bb		0.033		0.032		0.067
	(Ba − Bb)/(Ba + Bb)		0.020		0.076		0.264
CD5	Ba	617.40–656.03	0.498	576.46–614.83	0.471	576.46–614.83	0.461
	Bb	658.62–697.50	−0.396	535.79–573.91	−0.411	826.23–866.15	−0.405
	Ba − Bb		0.412		0.459		0.532
	Ba/Bb		−0.498		0.023		0.048
	(Ba − Bb)/(Ba + Bb)		−0.499		−0.060		0.327
CD6	Ba	658.62–749.25	0.414	415.44–492.89	0.235	658.62–739.25	0.375
	Bb	576.46–656.03	−0.401	410.49–412.97	−0.196	576.46–656.03	−0.344
	Ba − Bb		0.407		0.232		0.360
	Ba/Bb		−0.119		0.013		−0.417
	(Ba − Bb)/(Ba + Bb)		−0.015		0.151		−0.368

Note: Origin—Spectral information of reflectance spectra subjected to mean filtering, envelope removal, and first-order differentiation transformation. CD1–CD6—1^st^- to 6^th^-order wavelet high-frequency components on the transformed spectral basis. Ba and Bb—The characteristic spectral bands on which the correlation coefficients with Comprehensive Score for Photosynthetic System Chilling Injury (CSPC) are maximum and minimum (maximum negative correlation, excluding mean-filtered origin).

**Table 6 plants-12-03138-t006:** Information for each chilling treatment group.

Treatment Temperature (Daily Maximum Temperature/Daily Minimum Temperature)	Duration of Treatment	Marker	Batch of Treatment (Start Date–End Date)
25 °C/15 °C (Control temperature)	3 d	CKD3	14/12/2021–17/12/2021, 15/12/2021–18/12/2021, 16/12/2021–19/12/2021 10/12/2022–13/12/2022, 11/12/2022–14/12/2022, 12/12/2022–15/12/2022
	6 d	CKD6	14/12/2021–20/12/2021, 15/12/2021–21/12/2021, 16/12/2021–22/12/2021 10/12/2022–16/12/2022, 11/12/2022–17/12/2022, 12/12/2022–18/12/2022
	9 d	CKD9	14/12/2021–23/12/2021, 15/12/2021–24/12/2021, 16/12/2021–25/12/2021 10/12/2022–19/12/2022, 11/12/2022–20/12/2022, 12/12/2022–21/12/2022
20 °C/10 °C (Slightly chilling)	3 d	T1D3	23/12/2021–26/12/2021, 24/12/2021–27/12/2021, 25/12/2021–28/12/2021 19/12/2022–22/12/2022, 20/12/2022–23/12/2022, 21/12/2022–24/12/2022
	6 d	T1D6	23/12/2021–29/12/2021, 24/12/2021–30/12/2021, 25/12/2021–31/12/2021 19/12/2022–25/12/2022, 20/12/2022–26/12/2022, 21/12/2022–27/12/2022
	9 d	T1D9	23/12/2021–1/1/2022, 24/12/2021–2/1/2022, 25/12/2021–3/1/2022 19/12/2022–28/12/2022, 20/12/2022–29/12/2022, 21/12/2022–30/12/2022
15 °C/5 °C (Moderate chilling)	3 d	T2D3	23/12/2021–26/12/2021, 24/12/2021–27/12/2021, 25/12/2021–28/12/2021 19/12/2022–22/12/2022, 20/12/2022–23/12/2022, 21/12/2022–24/12/2022
	6 d	T2D6	23/12/2021–29/12/2021, 24/12/2021–30/12/2021, 25/12/2021–31/12/2021 19/12/2022–25/12/2022, 20/12/2022–26/12/2022, 21/12/2022–27/12/2022
	9 d	T2D9	23/12/2021–1/1/2022, 24/12/2021–2/1/2022, 25/12/2021–3/1/2022 19/12/2022–28/12/2022, 20/12/2022–29/12/2022, 21/12/2022–30/12/2022
10 °C/0 °C (Severe chilling)	3 d	T3D3	14/12/2021–17/12/2021, 15/12/2021–18/12/2021, 16/12/2021–19/12/2021 10/12/2022–13/12/2022, 11/12/2022–14/12/2022, 12/12/2022–15/12/2022
	6 d	T3D6	14/12/2021–20/12/2021, 15/12/2021–21/12/2021, 16/12/2021–22/12/2021 10/12/2022–16/12/2022, 11/12/2022–17/12/2022, 12/12/2022–18/12/2022
	9 d	T3D9	14/12/2021–23/12/2021, 15/12/2021–24/12/2021, 16/12/2021–25/12/2021 10/12/2022–19/12/2022, 11/12/2022–20/12/2022, 12/12/2022–21/12/2022

Note: One strawberry plant per group was treated in each treatment batch. The markers CK, T1, T2, and T3 correspond to temperature conditions of 25/15 °C, 20/10 °C, 15/5 °C, and 10/0 °C, respectively. The labels D3, D6, and D9 denote treatment durations of 3, 6, and 9 days, respectively.

## Data Availability

The original contributions presented in the study are included in the article. Further inquiries can be directed to the corresponding author.

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
