# Peer review of "Quantifying Chilling Injury on the Photosynthesis System of Strawberries: Insights from Photosynthetic Fluorescence Characteristics and Hyperspectral Inversion"

_plants, 2023, doi:10.3390/plants12173138_

Round 1

Reviewer 1 Report

Dear authors! The assessment of the impact of cold damage on the plant photosynthesis system is very important from a scientific and practical point of view. This determines the relevance of the study.

Despite this, there are a few remarks.

1. the absence of a control variant makes it difficult to objectively evaluate the results;

2. the results of biometric measurements of studies are not presented;

3. there is no detailed description of the crop cultivation technology;

4. The structure of the article does not fully comply with IMRAD standards. Section 4 (methods) should be inserted after the introduction;

5. Strawberry seedlings were purchased from a third party seller, so the authors had no control over their identity.

Author Response

Response to Reviewer 1 Comments

Manuscript ID: plants-2582278

Esteemed reviewer:

Greetings!

We sincerely appreciate your careful review of our manuscript. Your professional and meticulous suggestions have been carefully considered and addressed. Your valuable comments have helped us to improve the manuscript.

In this cover letter, we provide explanations to the main issues raised by you. Other issues such as formatting, typos, and grammar errors have been detailed in the revised manuscript used the

“Track Changes” in the attached file (which includes revisions made according to all the reviewers' comments). We have highlighted the main issues you raised with comments.

We express our gratitude for your hard work on our manuscript. It is our pleasure to work with you. If there are any further areas for improvement, we will continue to work to make our contribution to plant science.

Wish you a successful career and a happy life!

Best regards~

Sincerely yours, Author of plants-2582278

Point 1: The absence of a control variant makes it difficult to objectively evaluate the results.

Response 1: Thank you for your suggestion. The 25/15°C treatment serves as the control group (CK) in our study. Perhaps due to our incorporation of temperature variations on a daily basis, diurnal temperature difference, and duration into the CK design, it may appear that the control group resembles a treatment group. Regarding the CK design, our considerations were as follows:

(1) The 0/10°C, 5/15°C, and 10/20°C treatments with intervals of 5°C among the three low-temperature treatment groups allow for a comprehensive study of the effects of above-zero low temperatures on strawberries. The 10°C diurnal temperature difference is based on the actual temperature trend in greenhouse and horticultural production. The temperature range of 15~25°C is widely acknowledged as the optimal growth range for strawberries. While maintaining the ideal growth temperature, the 10°C temperature difference remains consistent with the low-temperature treatment groups, thus preserving the ideal growth conditions for strawberries and minimizing potential additional effects on strawberry physiological characteristics due to varying temperature differences.

(2) During the temperature-controlled experiments of 3, 6, and 9 days, strawberry plants are continuously growing. We cannot ascertain whether the photosynthetic physiological characteristics of strawberries may change with ongoing growth and development during this period from 3 to 9 days. Therefore, we applied the same 3, 6, and 9-day treatment to the CK, similar to the low-temperature treatment groups.

In summary, these considerations may make the CK of our study appear to resemble the fourth treatment group, while it is, in fact, the control group. We hope our explanation addresses your concerns.

Point 2: The results of biometric measurements of studies are not presented.

Response 2: Thank you for your suggestion! Indeed, this paper did not employ tables to present the mean (Mean), standard deviation (SD), and significant differences of various strawberry photosynthetic physiological parameters. However, in Figures 1 to 8, the paper meticulously showcases the mean and standard deviation of each observed parameter, employing bar graphs and error bars to illustrate the biostatistical aspects. Furthermore, we have modified the figure captions to read 'Each value is presented as "mean ± standard deviation (SD)" in the figure.' to alleviate any misunderstanding. We have opted for visual representation through images rather than tables due to the following reasons:

(1) The combination of temperature and time results in twelve treatment variables, in addition to the two strawberry varieties. Presenting this data in tabular form would consume a significant portion of the layout, potentially hindering optimal space utilization and the clarity of visual representation for readers' comprehension.

(2) In the textual descriptions of the 'Results' section, specific numerical details for key physiological parameters' statistical results and intergroup differences are enumerated, thus minimizing redundant presentation of insignificant variance data.

(3) Bar graphs offer a more intuitive way to illustrate differences between varieties and changes in strawberry physiological indicators under varying temperature and time conditions compared to tabular data.

We hope our explanation satisfactorily addresses your concerns.

Point 3: There is no detailed description of the crop cultivation technology.

Response 3: Thank you for your guidance! In accordance with your suggestion, we have supplemented the "Discussion" section with content relevant to the "Cultivation" aspect of this special issue. This section is based on the conclusions drawn from the preceding theoretical research, offering specific guidance for future strawberry cultivation in terms of quantitative and non-contact diagnosis of chilling injury to the photosynthetic system. The additional content is as follows: "Based on the aforementioned theoretical research findings, we posit that for the future practical cultivation of strawberries, it is feasible to deploy hyperspectral measuring devices covering the wavelength ranges of 576.46-614.83nm, 826.23-866.15nm, and 882.18-908.99nm above the plant canopy. By integrating machine learning algorithm models, this approach could facilitate rapid, quantitative, non-contact monitoring of chilling injury to the photosynthetic system of strawberries."

Point 4: The structure of the article does not fully comply with IMRAD standards. Section 4 (methods) should be inserted after the introduction.

Response 4: We also share your viewpoint. It is indeed more aligned with the logical flow of reading to place the "Methods" section after the "Introduction." However, the formatting requirements of this "Plants" journal stipulate that the "Methods" section be situated in the fourth section. Consequently, the sequence of these sections might not be within our purview to modify, as it must adhere to the layout prescribed by the journal. We appreciate your valuable insights!

Point 5: Strawberry seedlings were purchased from a third party seller, so the authors had no control over their identity.

Response 5: We would like to explain that the strawberry plants used in our two-year, two-batch experiments were sourced from the same strawberry cultivation facility. We have an established collaboration with this vendor, and for the two strawberry varieties mentioned in our paper, we made advance arrangements with the strawberry facility for their cultivation. The quality of the seedlings and the cultivation techniques are reliable. As such, we possess a certain level of familiarity with the identity of the strawberry vendor and can ensure the similarity in strawberry seedling quality for each experimental batch. We hope this explanation satisfies your inquiry. We appreciate your valuable input!

Reviewer 2 Report

The manuscript describes in detail the use of various physiological and optical methods to evaluate the susceptibility of selected strawberry varieties. Due to the broad approach and the multitude of methods used, it deserves to be published in “Plants”.

However minor corrections are necessary, as listed in a separate file.

Some minor corrections are needed.

Author Response

Response to Reviewer 2 Comments

Manuscript ID: plants-2582278

Esteemed reviewer:

Greetings!

We sincerely appreciate your careful review of our manuscript. Your professional and meticulous suggestions have been carefully considered and addressed. Your valuable comments have helped us to improve the manuscript.

In this cover letter, we provide explanations to the main issues raised by you. Other issues such as formatting, typos, and grammar errors have been detailed in the revised manuscript used the

“Track Changes” in the attached file (which includes revisions made according to all the reviewers' comments). We have highlighted the main issues you raised with comments.

We express our gratitude for your hard work on our manuscript. It is our pleasure to work with you. If there are any further areas for improvement, we will continue to work to make our contribution to plant science.

Wish you a successful career and a happy life!

Best regards~

Sincerely yours, Author of plants-2582278

【Tittle

Point 1: Title is too long: 22 words. I recommend to shorten the title to maximum 18 words, preferably nomore than 15. “Keywords” are the site where the Authors can include names of specific methods.

 Response 1: Based on your recommendations, we have removed some adjectives or phrases in the title. Additionally, we have supplemented the keywords accordingly, resulting in shortening the title to 17 words. The revised title is "Quantifying Chilling Injury on the Photosynthesis System of Strawberries: Insights from Photosynthetic Fluorescence Characteristics and Hyperspectral Inversion".

【Keywords

Point 2: Should not repeat words included in the Title, should be listed in alphabetical order.

Response 2: Following your suggestions, we have revised the keywords to avoid repetition with the title. Additionally, the keywords have been rearranged in alphabetical order. The revised keywords are "Chlorophyll fluorescence; Different photoperiod types; Flowering and fruit-setting stage; Fragaria × ananassa Duch; Gas exchange parameters; Principal component analysis; SIF; Spectral index; Sustained low temperature; Wavelet transform".

【Lines 64-65】

Point 3: Please use the conventional symbols of gas exchange (gs) transpiration (E) etc., use subscripts when necessary (see, for example LI-6400 manual).

Response 3: Following your advice, we have replaced the original terms 'Pn, Gs, Tr, Ci/Ca' with the more standardized abbreviations 'A, gs, E, ci/ca' in both the text and figures, respectively. We appreciate your guidance.

【Lines 166-167, 221-222】

Point 4: Figure captions please correct according to English language style.

Response 4: Thank you for your guidance. We have made revisions to all figures with non-standard expressions like: Panel (1a), (1c), and (1e) represent the maximum net photosynthetic rate (Amax), light compensation point (LCP), and light saturation point (LSP) of the short-day strawberry cultivar 'Toyonoka', respectively. Panel (1b), (1d), and (1f) correspondingly illustrate the Amax, LCP, and LSP of the long-day cultivar 'Selva'.

【Lines 162, 217, 258 etc】

Point 5: ALL Figures, particularly 1-8: The texts and numbers/letters inside the

figures are to some extend illegible, I recommend to enlarge the figures inserted into the text and preferably with enlarging the smaller fonts.

Response 5: In accordance with your suggestion, we have adjusted the size of numbers and characters in all the figures throughout the manuscript. As a result, the readability of the figures has been enhanced. A comparison between the figures before and after the adjustments is presented in the attached file.

【Line 165】

Point 6: Figure 1, leaf chlorophyll content and leaf soluble protein content were not described in the Materials and Methods chapter. Please insert the proper data.

Response 6: Thank you for your correction! We deeply regret such a basic mistake occurring in our work. The terms 'leaf chlorophyll content' and 'leaf soluble protein content' were not relevant in this paper. 'LCP' and 'LSP' correspond to the abbreviations of 'light compensation point' and 'light saturation point,' respectively.

【Line 234 & 244】

Point 7: WUE is calculated as a ratio of Pn and transpiration (as mentioned in M&M chapter) so the units should be expressed consequently as μmol/mmol, unless the authors converted moles to grams, but taking into account the values shown in the Figures and in the text they did not. Please explain the parameter AQE and give the units, the reference cited as [52] in M&M chapter does not describe this parameter, please apply another references using this parameter.

Response 7: Thank you for your correction! We have, in accordance with your suggestion, revised the units of WUE throughout the manuscript and figures from g Kg-1 to μmol mmol-1. Furthermore, we have provided an explanation for AQE as follows: "The apparent quantum efficiency (AQE) characterizes the efficiency of photosynthetic conversion of light energy into biomass energy in plants." Additionally, we have included the unit of AQE, "", both in the text and within the figures. The reference for the AQE calculation method has now been added as reference '53'.

【Others】

Point 8: Details and grammatical issues

Response 8: In addition to addressing the main issues as mentioned above, we have thoroughly reviewed the entire manuscript once again. Furthermore, based on your suggestions, we have italicized the Latin name of strawberry in 【Line 35】, provided clarification for the meaning of "Level" in 【Line 52】, corrected the spelling of "quantum yield" in 【Line 381】, refined the description of the container specifications in 【Line 858】, rectified the numbering of Figure 15, and revised the title of Section 4.4.2.
